# Defining latent spaces by example: optimisation over the outputs of generative models

## Abstract

Modern generative AI models like diffusion and flow matching can sample from rich data distributions, but many downstream tasks — such as experimental design or creative content generation — require a higher level of control than unconstrained sampling. Here, the challenge is to efficiently identify outputs that are both probable under the model and satisfy task-specific constraints. Often, the evaluation of samples is expensive and lack gradients — a setting known as black-box optimisation. In this work, we allow black-box optimisation on top of diffusion and flow matching models for the first time by introducing *surrogate latent spaces*: non-parametric, low-dimensional Euclidean embeddings that can be extracted from any generative model without additional training. The axes can be defined via examples, providing a simple and interpretable approach to define custom latent spaces that express intended features and is convenient to use in downstream tasks. Our proposed representation is Euclidean and has controllable dimensionality, permitting direct application of standard optimisation algorithms. We demonstrate that our approach is architecture-agnostic, incurs almost no additional computational cost over standard generation, and generalises across modalities, including images, audio, videos, and structured objects like proteins.

## 1 Introduction

In recent years, generative models have undergone a step-change in performance. Whereas a decade ago they were often tailor-made and domain-specific, modern sample-based approaches such as diffusion and flow matching provide a general framework. Their strength lies in avoiding explicit modelling of the data-generating process and instead specifying a transformation between data samples and a latent distribution. This paper focuses on achieving the fine-grained control of latent variables ("the noise") required to encourage desired output properties — a strategy known as Latent Space Optimisation (LSO) (Gómez-Bombarelli et al., 2018). LSO is often simpler and more efficient than directly searching in the data space, as the model may encode structural details of the data, resulting in a latent space with simpler structure than the original data manifold.

This paper introduces a general and model-agnostic framework for black-box optimisation over the outputs of diffusion and flow-matching models. Unlike previous methods this allows the optimisation of expensive, and/or gradient-free objectives relevant in many areas of experimental design. The key idea is to construct explicit low-dimensional approximately Euclidean spaces whose coordinates are defined by examples and which map bijectively to valid model outputs. Under deterministic generation — e.g. flow matching (Lipman et al., 2022), DDIM (Song et al., 2020a), or probability-flow ODE sampling (Song et al., 2020b)) — these surrogate spaces allow any optimiser to operate directly over model outputs — providing a practical optimisation capability that was previously unavailable for sample-based generative models.

While LSO has been applied successfully in the context of Variational Autoencoders (VAE) (Gómez-Bombarelli et al., 2018; Kusner et al., 2017), there has been limited progress in applying LSO to sample-based models such as diffusion and normalising flows due to two key challenges. Challenge 1 (C1): the latent variable in these models retain the dimensionality of their generated outputs, and so efficient exploration is hampered by the curse of dimensionality. Challenge 2 (C2): until recently it was unclear how to safely manipulate the latent variable without leaving the support of the model, yielding unrealistic generations. Moreover, although alternative optimisation algorithms

have been designed for sample-based models (Krishnamoorthy et al., 2023; Li et al., 2025), they are sophisticated, model-specific, and do not operate on the latent variable.

In this paper we show how the challenges C1 and C2 can be circumvented, enabling a wide range of downstream optimisation applications to benefit from the generative performance of sample-based models. In particular, we propose by-example-specified latent spaces which let us construct targeted latent spaces with a dimensionality of our choosing *without* losing generation fidelity, e.g. Figure 1 shows a 2-dimensional space constructed from three examples. Our proposed framework builds upon recent work (Bodin et al., 2024) for building low-dimensional subspaces of the latent distribution and has the following key takeaways:

1. We provide a practical mechanism enabling black-box optimisation over the outputs of diffusion and flow-matching models.

2. We introduce *surrogate latent spaces* using a coordinate chart from the Euclidean space to a latent subspace. This allows us to form simple-to-use subspaces which are completely customisable by choosing the "seed latents" which defines the space.

3. We show that our surrogate latent spaces are well-suited for popular optimisation methods, including Bayesian Optimisation (BO) and CMA-ES, opening the door for LSO using diffusion and flow matching models.

4. Our approach is general and can be applied to any latent variable model, such as a pre-trained diffusion or flow matching model, and across a range of data modalities (see Figure 2). In a key experiment we take a state-of-the-art protein generation framework and show a significant improvement in the number of successful generations, allowing us to generate proteins of a greater length than was previously feasible.

## 2 BACKGROUND

**Deterministic generation in diffusion and flow matching models**. Both diffusion (Ho et al., 2020; Song et al., 2020b;a) and flow matching (Lipman et al., 2022) models have the ability to perform *deterministic generation*. In this setting a latent variable $z \sim p$ — where $p$ is the latent distribution — fully specifies the generated data $x$. Moreover, using reverse generation procedures, they allow for a known data object $x$ to be reverted to its corresponding latent representation $z$ — a process referred to as *inversion*. Determinism creates a mapping between the latent space and the data space, which is a key feature we exploit in our optimisation approach. With stochastic generation the latent variable would only be weakly informative, substantially reducing the possibility for controlled generation.

**Black-box optimisation** targets problems where the objective function is explicitly unknown *a-priori* but can be queried through function evaluations. These evaluations may be expensive, noisy, or derivative-free, and given that the function is not directly accessible such problems are referred to as "black-box". In such settings optimisation techniques such as Bayesian Optimisation (BO) (Shahriari et al., 2015), Evolutionary Strategies (ES) (Hansen and Ostermeier, 2001), and Particle-Swarm Optimisation algorithms (PSO) (Kennedy and Eberhart, 1995) are commonly employed.

A typical black-box optimisation problem can be written as $x^* = \arg\max_{x \in \mathcal{X}} f(x)$, where $f: \mathcal{X} \to \mathbb{R}$ is the objective function, and $\mathcal{X}$ is the domain of interest. In our setting, this domain may be for example possible protein structures or possible images, and $f$ evaluates a property of the generated sample (e.g., biological fitness of a protein structure or aesthetic quality of an image).

**Black-box optimisation in the latent space of generative models**. An increasingly common setting for black-box optimisation is searching over the latent space $\mathcal{Z}$ associated with a generative model, i.e. Latent Space Optimisation(LSO) (Kusner et al., 2017; Gómez-Bombarelli et al., 2018; Lu et al., 2018; Luo et al., 2018), which seeks to solve

$$z^* = \arg\max_{z \in \mathcal{Z}} f(g(z)),$$

where $g: \mathcal{Z} \to \mathcal{X}$ is a generative model, and $f$ the black-box objective defined on generated objects in $\mathcal{X}$. LSO was first popularised by Gómez-Bombarelli et al. (2018) in the context of molecule design, where latent representations from VAEs were used to facilitate gradient-based optimisation of drug candidates.

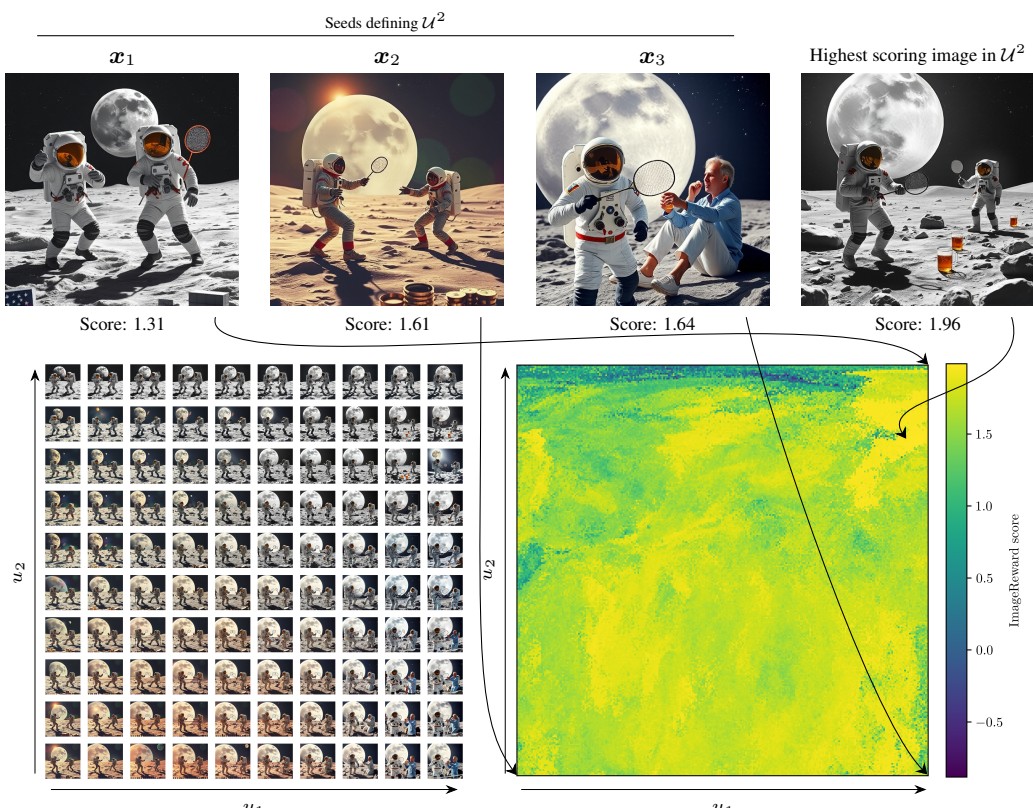

Figure 1: **Surrogate latent spaces**. (*bottom left*) Generations associated with a grid over a 2-dimensional surrogate latent space, formed using the latent vectors corresponding to the examples $x_1$, $x_2$, and $x_3$ for the FLUX.1-schnell (Labs, 2024) rectified flow model. (*bottom right*) The ImageReward score for a target prompt (the objective function) over a dense grid ($256 \times 256$) of generations from our surrogate space show rich structure that can be exploited by standard optimisers. The example ('seed') images were obtained by sampling the model using the target prompt but they fail to follow it; by navigating our surrogate space, we can find images with better alignment.

## 3 SURROGATE LATENT SPACES

We now present our approach for effective LSO in modern deep generative models. The proposed approach derives a low-dimensional *surrogate latent space* from a pre-trained generative model such that optimisation methods can effectively be applied regardless of the model's dimensionality.

**Summary.** In Figure 3 we illustrate at a high level how surrogate spaces are defined and work. Using a set of $K$ freely chosen examples, encoded as realisations of the latent variable $z_1, \dots, z_K \sim p$ from the latent distribution $p$, a $(K-1)$-dimensional bounded space $\mathcal{U} = [0,1]^{K-1}$ will be defined; this is a surrogate latent space. Each point $u \in \mathcal{U}$ maps to an unique weight vector $w$, which in turn maps to an unique latent realisation $z \sim p$, and in turn to a generated object $x$ by the generative model. The seeds form a coordinate system of latent realisations — implying a coordinate system of objects that can be generated. We now introduce the guiding principles by which these spaces are constructed.

Surrogate latent spaces are designed to follow three key principles, each supporting the straightforward application of popular black-box optimisation algorithms described above:

P1 **Validity**: All locations (in the surrogate space) must be supported by the generative model.

P2 **Uniqueness**: All locations must encode unique objects.

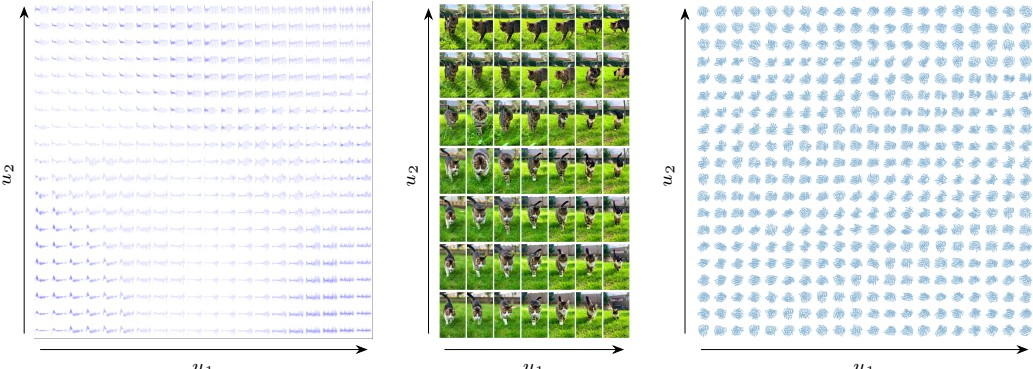

Figure 2: **We can build smooth surrogate spaces for any generative model.** (*left*) Waveform generations over a grid of a 2D slice of a 7D surrogate space formed from 8 seed latents and the 8256-dimensional StableAudio2.0 text-to-audio generation model (Evans et al., 2025). (*middle*) The first frames of a similarly constructed grid of videos from the 4,308,480-dimensional HunyuanVideo text-to-video generation model (Kong et al., 2024). (*right*) A grid of proteins over a 2D surrogate space formed from 3 seed latents corresponding to 3 proteins using RFDiffusion (Watson et al., 2023).

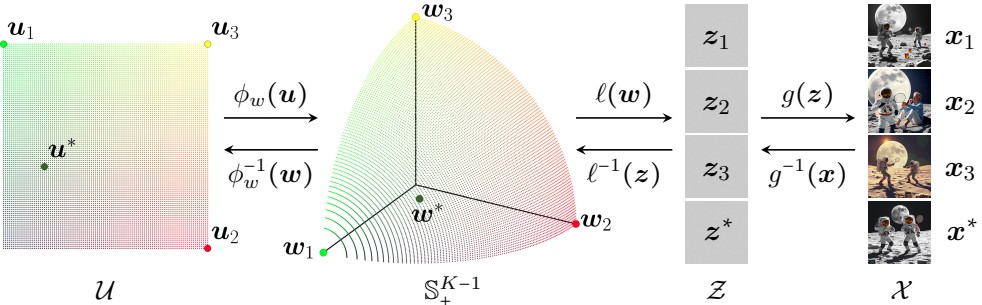

Figure 3: **Illustration of a surrogate latent space.** *(left)* A low-dimensional surrogate space $\mathcal{U}$ is mapped via the surrogate chart $\phi_w$ onto the positive orthant of the unit sphere *(mid left)*, which is then transported via $l$ to valid latent realisations $\mathbf{z} \sim p$ *(mid right)*, before finally being decoded by the generative model $g$ into objects $\mathbf{x}$ *(right)*. All mappings are bijective, and enable optimisation algorithms to operate in $\mathcal{U}$ while guaranteeing validity, uniqueness, and approximate stationarity.

P3 **Stationarity**: The relationship between objects' similarity as a function of their Euclidean distance in the surrogate space should be approximately maintained for any pair of objects throughout the space.

The combination of these three principles ensures that the optimisation algorithm can remain agnostic to the generative model. P1 allows the algorithm to safely navigate the space without exploring areas which yield invalid or undefined inputs to the generative model, producing invalid generations. P2 avoids redundancy, ensuring that locations treated as distinct by the algorithm do not encode the same solution. P3 helps the optimisation algorithm by avoiding substantial warpings of the space — where the generated object may change rapidly in some areas and very slowly in some areas.

**Defining $\mathcal{U}$** The key idea behind our proposed method is to build a search space $\mathcal{U}$ using a collection of $K$ so-called *seeds*, resulting in a coordinate system able to generate objects with similar properties to the seeds. The surrogate and the latent space are related through a smooth bijective mapping $\phi : \mathcal{U}, \cdot \to \mathcal{Z}$, which we refer to as the *surrogate chart*. The surrogate chart is associated with the seed latents $\{\mathbf{z}_k\}_{k=1}^K$, $\mathbf{z}_k \in \mathcal{Z}$ and defined as,

$$\phi(\mathbf{u}, \{\mathbf{z}_k\}_{k=1}^K) = \mathbf{z}, \qquad \mathbf{z} := l\big(\mathbf{w}, \{\mathbf{z}_k\}_{k=1}^K\big), \qquad \mathbf{w} := \phi_w(\mathbf{u}), \qquad (1)$$

where $\phi_w : \mathcal{U} \to \mathbb{S}_+^{K-1}$ and $l : \mathbb{S}_+^{K-1}, \cdot \to \mathcal{Z}$ are invertible functions. Here, $\mathbb{S}_+^{K-1}$ denotes the positive orthant of the unit $(K-1)$-sphere,

$$\mathbb{S}_+^{K-1} := \left\{ \mathbf{w} \in \mathbb{R}^K \mid \|\mathbf{w}\|_2 = 1, \ w_i \geq 0 \ \forall i \right\}.$$

The surrogate chart $\phi$ thus defines a coordinate system for a subset of $\mathcal{Z}$, with inverse mapping

$$\phi^{-1}(\mathbf{z}, \{\mathbf{z}_k\}_{k=1}^K) = \mathbf{u}, \qquad \mathbf{u} := \phi_w^{-1}(\mathbf{w}), \qquad \mathbf{w} := l^{-1}\big(\mathbf{z}, \{\mathbf{z}_k\}_{k=1}^K\big). \qquad (2)$$

**Seed selection.** The seeds $\{\mathbf{z}_k\}$ specify the directions that define the surrogate chart, and therefore determine which $(K-1)$-dimensional manifold inside the full latent space is explored. The method itself imposes no requirement that these seeds be curated or high-scoring: any non-degenerate set of samples produces a valid surrogate space and a bijection between $u \in \mathcal{U}$ and the corresponding subset of the latent space. Empirically (Section 5), both selected seeds — e.g. inversions from known, high-scoring objects — and random seeds lead to surrogate spaces that support effective optimisation, though the attainable variation and the region targeted by the search depend on the seed set. Increasing $K$ expands the subset of latent space spanned by the seeds which generally increases the diversity of attainable generations, but it also raises the dimensionality of the surrogate space (which can make optimisation more challenging), introducing a natural trade-off. In practice, practitioners may use task-specific seeds when available, supplement them with random seeds when needed, or rely entirely on random seeds when no task-specific information exists.

### 3.1 Ensuring model support for all coordinates in $\mathcal{U}$

To ensure validity (P1), each coordinate $\mathbf{u} \in \mathcal{U}$ maps to a latent realisation $\mathbf{z} \in \mathcal{Z}$ via a linear combination weight vector $\mathbf{w} \in \mathbb{S}_+^{K-1}$. To guarantee that all coordinates $\mathbf{u} \in \mathcal{U}$ are valid, the weight vector $\mathbf{w}$ must map to a $\mathbf{z} \sim p$, where $p$ is the latent distribution of the generative model. To ensure this, the function $l$ (Equation 1) is formed via a *Latent Optimal Linear combinations (LOL)* transport map (Bodin et al., 2024) which via an 'inner' latent variable $\boldsymbol{\epsilon} \sim p_\epsilon$ guarantees that a linear combination of the seed latents follows the latent distribution $p$

$$\mathbf{z} = l\big(\mathbf{w}, \{\mathbf{z}_k\}_{k=1}^K\big) := \mathcal{T}_\leftarrow(\boldsymbol{\epsilon}) \qquad \boldsymbol{\epsilon} := \boldsymbol{\xi}\mathbf{w} \qquad \boldsymbol{\xi} := [\boldsymbol{\epsilon}_1, \ldots, \boldsymbol{\epsilon}_K]^T \qquad \boldsymbol{\epsilon}_k := \mathcal{T}_\rightarrow(\mathbf{z}_k), \quad (3)$$

where $\mathcal{T}_\rightarrow$ and $\mathcal{T}_\leftarrow$ maps from $p$ to $p_\epsilon$ and back, respectively. Points in $\mathcal{Z}$ which have been generated by the forward map $\mathbf{u} \mapsto \mathbf{w} \mapsto \mathbf{z}$ admits an exact inverse: $\mathbf{w} = l^{-1}(\mathbf{z}, \{\mathbf{z}_k\}_{k=1}^K) = \boldsymbol{\xi}^+ \mathcal{T}_\rightarrow(\mathbf{z})/\|\boldsymbol{\xi}^+ \mathcal{T}_\rightarrow(\mathbf{z})\|_2$ where $\boldsymbol{\xi}^+$ is the Moore–Penrose inverse of $\boldsymbol{\xi}$, see Section A. Here typically $dim(\boldsymbol{\epsilon}) = dim(\mathbf{z})$ although this in general will depend on the transport map.

**Inner latents closed under linear combinations** An inner latent distribution $p_\epsilon$ is amenable to our methodology if it is zero-mean, rotationally invariant, and closed under aggregation with unit–$\ell_2$ weights. Formally, if $\boldsymbol{\epsilon}_1, \ldots, \boldsymbol{\epsilon}_K \overset{\text{i.i.d.}}{\sim} p_\epsilon$ and $\mathbf{w} \in \mathbb{S}^{K-1}$, then

$$\mathbf{z} = \mathcal{T}_\leftarrow(\boldsymbol{\xi}\mathbf{w}), \qquad \boldsymbol{\xi} = [\boldsymbol{\epsilon}_1, \ldots, \boldsymbol{\epsilon}_K]^T, \qquad (4)$$

again satisfies $\mathbf{z} \sim p$. Specifying the maps $T_\rightarrow$ and $\mathcal{T}_\leftarrow$ to map between the distributions $p$ and $p_\epsilon$ is the framework for applying our method to a model at hand:

- **Gaussian latents.** If $p = \mathcal{N}(\mathbf{0}, \boldsymbol{\Sigma})$, then closure holds directly under $\|\mathbf{w}\|_2 = 1$, and we may set $\mathcal{T}_\rightarrow = \mathcal{T}_\leftarrow = \text{id}$. If $p = \mathcal{N}(\boldsymbol{\mu}, \boldsymbol{\Sigma})$ with $\boldsymbol{\mu} \neq \mathbf{0}$, then $\mathcal{T}_\rightarrow(\mathbf{z}) = \mathbf{z} - \boldsymbol{\mu}$ and $\mathcal{T}_\leftarrow(\boldsymbol{\epsilon}) = \boldsymbol{\epsilon} + \boldsymbol{\mu}$, which centres the distribution before aggregation and restores the mean afterwards. This case was treated in Bodin et al. (2024).

- **Hyperspherical latents.** If $p = \text{Unif}(\mathbb{S}^{D-1})$, where $\mathbf{z} \in \mathbb{R}^D$, then $\mathcal{T}_\rightarrow = \text{id}$, while $\mathcal{T}_\leftarrow$ normalises any linear combination back onto the sphere, $\mathcal{T}_\leftarrow(\boldsymbol{\epsilon}) = \frac{\boldsymbol{\epsilon}}{\|\boldsymbol{\epsilon}\|}$. Because this construction is rotation-equivariant, it preserves the uniform law on the sphere.

- **Composite latents.** If the latent variable $\mathbf{z}$ can be decomposed into $M$ statistically independent components such that $\mathbf{z} = \{\mathbf{z}^{(1)}, \ldots, \mathbf{z}^{(M)}\}$ where $p(\mathbf{z}) \propto p(\mathbf{z}^{(1)}) \ldots p(\mathbf{z}^{(M)})$, then each component $\mathbf{z}^{(m)}, m \in [1, \ldots, M]$ can be mapped separately. This allows, for example, for a model having two or more latent variables, to map these independently, and concatenate their inner latents when computing linear combinations.

- **General scalar distributions** If a latent variable $z$ has individual dimensions $z_i$ that are independent of the others (see above), those elements follow scalar distributions which can be transported optimally to e.g. $\mathcal{N}(0,1)$ — which is amenable — using the respective cumulative distribution function as proposed in Bodin et al. (2024).

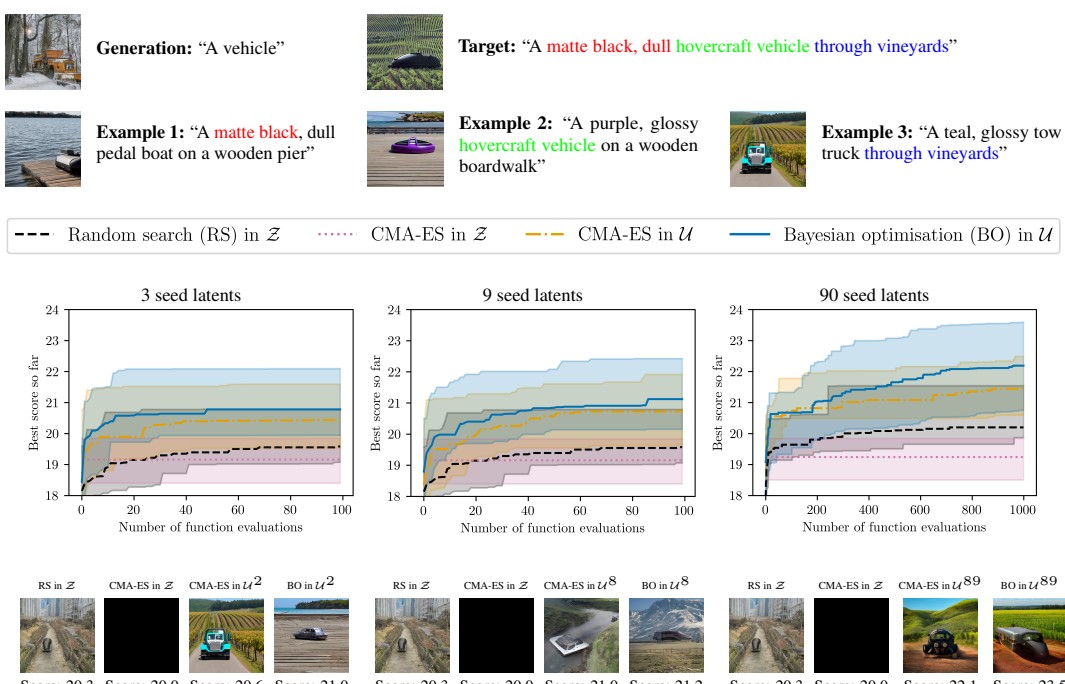

Figure 4: **Image optimisation given partially prompt fulfilling examples** Shown is the median and 90% confidence interval of the best-so-far score found per step across runs. The prompt given to the generative model (see Generation) and a sampled prompt given to the scorer (see Target) is shown in the top, together with example images (generated from the example prompts) used as seeds to form one of the surrogate space searched within. Each run is given its own target and seed examples (but the same across methods), and the number of examples is 3, 9 and 90 in its respective column, forming 2D, 8D and 89D surrogate spaces, respectively.

## 3.2 Ensuring uniqueness for generations from $\mathcal{U}$

We now address principle (P2) — that every point in $\mathcal{U}$ must specify a unique realisation of $z$. In Bodin et al. (2024) linear combinations of the seeds are defined for the entire hyperplane, however, following the transformation, such combinations can all be indexed by a bounded set; the weights $w \in \mathbb{S}^{K-1}$ is a sufficient such set to index all possible Latent Optimal Linear combinations, as we show in Section B. As each coordinate $u \in \mathcal{U}$ specifies a unique $w \in \mathbb{S}_+^{K-1}$, which in turn specifies a unique $z$, it follows that each coordinate in $\mathcal{U}$ maps to an unique latent realisation.

The focus on linear combinations weights residing on the positive orthant $\mathbb{S}_+^{K-1} \subset \mathbb{S}^{K-1}$ rather than the whole hypersphere will be motivated by the principle addressed in the next section. But we point out that, to represent positive associations to the seeds we will only need positive weights $\mathbb{S}_+^{K-1}$, as only those induce (positive) similarity to the seeds. Negative associations (to encourage dissimilarity) to any particular seed could still be represented, by negating the corresponding seed latent. Moreover, as $\mathbb{S}_+^{K-1}$ is a subset of $\mathbb{S}^{K-1}$, uniqueness still holds.

## 3.3 Ensuring approximate object similarity stationarity

We will now address our final principle of stationarity (P3), motivated by a well-established observation in generative modelling that similarity between generated objects is captured by the *cosine similarity* of latent vectors. This principle underlies embedding models (Devlin et al., 2019; Mikolov

et al., 2013; Kingma and Welling, 2013), widely in e.g. information retrieval (Hambarde and Proenca, 2023) and model alignment (Radford et al., 2021). We adopt the same assumption in this work.

A common implicit assumption when computing cosine similarities is that latent vectors are centred and isotropic, e.g., unit Gaussian (Steck et al., 2024). While this may not hold for general latent distributions, it does hold for the *inner latent* representation $\epsilon$ (Equation 3). Hence, $\epsilon$ serves as the central latent variable in this section.

We adopt $\mathrm{sim}_z(\boldsymbol{z}_i, \boldsymbol{z}_j) := \mathrm{sim}_e(\boldsymbol{\epsilon}_i, \boldsymbol{\epsilon}_j)$ where $\mathrm{sim}_e$ is the cosine similarity, which is

$$\mathrm{sim}_z(\boldsymbol{z}_i, \boldsymbol{z}_j) = \mathrm{sim}_e(\boldsymbol{\epsilon}_i, \boldsymbol{\epsilon}_j) = \frac{\boldsymbol{\epsilon}_i^\top \boldsymbol{\epsilon}_j}{\|\boldsymbol{\epsilon}_i\| \, \|\boldsymbol{\epsilon}_j\|} = \frac{(\boldsymbol{\xi} \boldsymbol{w}_i)^\top \boldsymbol{\xi} \boldsymbol{w}_j}{\|\boldsymbol{\xi} \boldsymbol{w}_i\| \, \|\boldsymbol{\xi} \boldsymbol{w}_j\|} \tag{5}$$

for $\boldsymbol{z}_i = \phi(\boldsymbol{u}_i)$ and $\boldsymbol{z}_j = \phi(\boldsymbol{u}_j)$, where $\boldsymbol{u}_i, \boldsymbol{u}_j \in \mathcal{U}$. For large latent dimensionality $D$, this reduces to

$$\frac{(\boldsymbol{\xi} \boldsymbol{w}_i)^\top \boldsymbol{\xi} \boldsymbol{w}_j}{\sqrt{(\boldsymbol{w}_i^\top \boldsymbol{\xi}^\top \boldsymbol{\xi} \, \boldsymbol{w}_i)(\boldsymbol{w}_j^\top \boldsymbol{\xi}^\top \boldsymbol{\xi} \, \boldsymbol{w}_j)}} \xrightarrow[D \to \infty]{\mathrm{a.s.}} \boldsymbol{w}_i^\top \boldsymbol{w}_j \tag{6}$$

since $\|\boldsymbol{w}_i\| = \|\boldsymbol{w}_j\| = 1$, $\mathbb{E}[\epsilon_{k,d}] = 0$, and $\boldsymbol{\xi}^\top \boldsymbol{\xi} \xrightarrow[D \to \infty]{\mathrm{a.s.}} D\sigma^2 \boldsymbol{I}$ for independent $\{\boldsymbol{\epsilon}_k\}$. In Section D we show that this effect dominates already at practical dimensionalities, allowing us to control the similarity of objects across $\mathcal{U}$ through the design of $\phi_w$.

To preserve similarity as a function of Euclidean distance, we require, for some function $v$

$$v(\|\mathbf{u}_i - \mathbf{u}_j\|_2) = \phi_w(\mathbf{u}_i)^\top \phi_w(\mathbf{u}_j), \quad \forall \, \mathbf{u}_i, \mathbf{u}_j \in \mathcal{U}, \tag{7}$$

the form of a stationary kernel. This condition can only hold approximately, since $\phi_w : [0,1]^{K-1} \to \mathbb{S}_+^{K-1}$ maps flat to curved space, and Gaussian curvature is preserved under local isometries; analogous to the impossibility of constructing a flat map of the globe that preserves all distances — the classic cartographic problem of Snyder (1987).

Restricting to an orthant reduces curvature and thus error, but exact preservation is unattainable for any $\phi_w$. In Appendix C, we specify two variants, including one based on the Knothe–Rosenblatt (KR) map (see Appendix E for analysis) which we found performs well empirically and is adopted in our experiments, unless specified otherwise.

## 4 RELATED WORK

**High-dimensional Bayesian Optimisation**. Alternative approaches to address the challenge of high-dimensional black-box optimisation include REMBO (Wang et al., 2016), sparse axis-aligned subspaces (Eriksson and Jankowiak, 2021), incumbent-guided subspaces (Ngo et al., 2025), and Turbo (Eriksson et al., 2019), all offering mechanisms for BO in high dimensionalities. However, these approaches have no way to stay in the support of diffusion and flow matching models. As such, they are complementary to our approach; specifically, our framework enables them to be deployed in the latent spaces of sample-based generative models by providing the Euclidean spaces these methods generally expect.

**Advances in LSO for VAEs.** There has been a recent focus on improving the sample efficiency of LSO by fine-tuning the VAE during optimisation (Grosnit et al., 2021; Tripp et al., 2020; Maus et al., 2022; Chu et al., 2024), or by otherwise constraining the generation process (Boyar and Takeuchi, 2024; Moss et al., 2025). However, the cost of repeated sampling or fine-tuning modern sample-based models makes these approaches prohibitively expensive for our setting.

**Alternative methods for optimisation with generative models.** A complementary line of work couples generative models with offline black-box optimisation by training conditional diffusion models over the function domain. These approaches require building a new generative model for each optimisation task, either through costly full training (Krishnamoorthy et al., 2023) or task-specific fine-tuning (Fan et al., 2023a; Denker et al., 2025). Because they rely on datasets of evaluated generations or on gradient information during sampling, they implicitly assume that the objective function is inexpensive to evaluate or has a known gradient, and that it is practical to train the generative model (which may be large). In contrast, our approach avoids any model training or

fine-tuning (see Table 1) and does not incur the additional computational cost or hyper-parameter sensitivity associated with directly steering the generative process — issues known to affect other inference-time techniques such as classifier or reconstruction guidance (Dhariwal and Nichol, 2021; Chung et al., 2023; Song et al., 2024).

| | "A green colored rabbit." | | "Two roses in a vase." | | "Two dogs in the park." | | GPU hrs (training) |
|---|---|---|---|---|---|---|---|
| | Reward (↑) | Diversity (↑) | Reward (↑) | Diversity (↑) | Reward (↑) | Diversity (↑) | |
| Standard model sampling | -0.16 [-0.35, 0.03] | 0.17 [0.15,0.18] | 0.80 [0.66, 0.90] | 0.12 [0.10, 0.13] | 0.36 [0.29, 0.43] | 0.16 [0.16, 0.17] | N/A |
| *1/2 efficiency* | | | | | | | |
| Best-1-of-2, 100 times | 0.54 [0.39, 0.73] | 0.17 [0.16, 0.18] | 1.22 [1.15, 1.30] | 0.09 [0.09, 0.10] | 0.63 [0.58, 0.69] | 0.16 [0.15, 0.16] | N/A |
| Best-100-of-200 | 1.01 [0.78, 1.17] | 0.17 [0.15, 0.18] | 1.40 [1.34, 1.45] | 0.09 [0.09, 0.10] | 0.73 [0.68, 0.78] | 0.15 [0.14, 0.16] | N/A |
| Grid in $\mathcal{U}^1$ (2 seeds) | 1.67 [1.08, 1.84] | 0.06 [0.03, 0.12] | 1.43 [0.70, 1.77] | 0.05 [0.04, 0.11] | 0.83 [0.32, 1.16] | 0.09 [0.07, 0.14] | N/A |
| Grid in $\mathcal{U}^3$ (4 seeds) | 1.55 [1.27, 1.77] | 0.08 [0.06, 0.13] | 1.29 [0.96, 1.53] | 0.08 [0.07, 0.11] | 0.74 [0.49, 0.93] | 0.13 [0.11, 0.15] | N/A |
| Grid in $\mathcal{U}^5$ (6 seeds) | 1.56 [1.28 1.74] | 0.09 [0.07, 0.13] | 1.34 [1.01, 1.58] | 0.09 [0.07, 0.12] | 0.68 [0.53, 0.83] | 0.13 [0.11, 0.15] | N/A |
| *1/6 efficiency* | | | | | | | |
| Best-1-of-6, 100 times | 1.42 [1.35, 1.50] | 0.13 [0.12, 0.14] | 1.55 [1.52, 1.58] | 0.09 [0.09, 0.10] | 0.91 [0.87, 0.95] | 0.15 [0.14, 0.15] | N/A |
| Best-100-of-600 | 1.65 [1.61, 1.69] | 0.11 [0.10, 0.12] | 1.64 [1.61, 1.65] | 0.09 [0.08, 0.10] | 1.00 [0.97, 1.03] | 0.15 [0.14, 0.15] | N/A |
| Grid in $\mathcal{U}^1$ (2 seeds) | 1.75 [1.50, 1.86] | 0.04 [0.03, 0.08] | 1.55 [1.19, 1.78] | 0.06 [0.04, 0.11] | 0.84 [0.35, 1.13] | 0.09 [0.07, 0.11] | N/A |
| Grid in $\mathcal{U}^3$ (4 seeds) | 1.73 [1.31, 1.83] | 0.07 [0.05, 0.08] | 1.40 [1.21, 1.65] | 0.08 [0.06,0.10] | 0.69 [0.32, 0.94] | 0.13 [0.11, 0.16] | N/A |
| Grid in $\mathcal{U}^5$ (6 seeds) | 1.71 [1.45, 1.79] | 0.07 [0.06, 0.09] | 1.32 [1.15, 1.55] | 0.09 [0.07, 0.12] | 0.68 [0.40, 0.86] | 0.14 [0.12, 0.17] | N/A |
| DPOK | 1.62 | 0.07 | 1.59 | 0.11 | 1.01 | 0.14 | 28 (A100) |
| Adjoint Matching | 1.71 | 0.05 | 1.50 | 0.09 | 1.33 | 0.13 | 4 (A100) |
| Importance Fine-tuning | 1.46 | 0.05 | 1.53 | 0.08 | 1.01 | 0.12 | 7 (RTX 4090) |

Table 1: **Scores and diversities of generated images** Reported is the median and the 90% confidence interval of the mean score and diversity, respectively, for each method (row) and prompt (shown in the top). Each section of rows correspond to methods using different compute budgets; with standard model sampling, 100 samples are drawn randomly from the (base) model (SD 1.5), while the bottom section are methods requiring training beyond the base model. The '1/2 efficiency' are methods with 100 initial random samples at its disposal before then producing the final 100 samples, and '1/6 efficiency' is the same but using 500 initial samples. The 'Best-of'-methods use the budget as per their name, and the 'Grid in $\mathcal{U}$'-methods forms surrogate spaces from seeds being the highest scoring samples among the initial random samples, followed by producing a grid of 100 points.

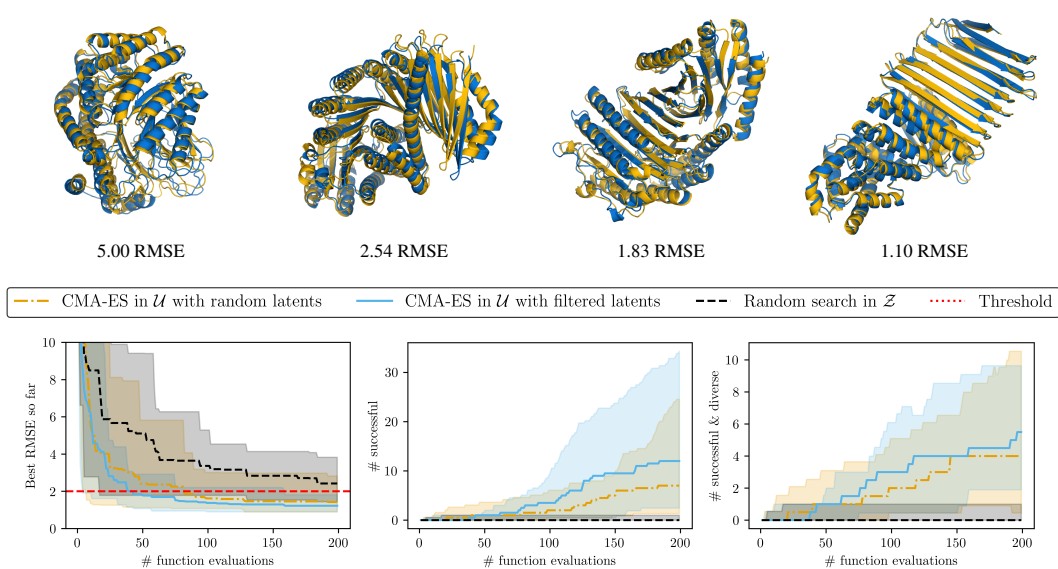

Figure 5: **Protein design with surrogate latent spaces.** *Top:* Representative generations showing the RMSE discrepancy between the RFDIFFUSION backbone *(yellow)* and their ALPHAFOLD2 regeneration *(blue)*. *Bottom:* comparison of standard sampling from the model versus optimisation in our surrogate spaces using CMA–ES. Plots report the median and 90% confidence interval of the best RMSE per step as well as the number of successful and diverse designs.

## 5 EXPERIMENTS

The experiments in Sections 5.1–5.3 are designed to demonstrate the key properties required for black-box optimisation over generative models: that the constructed surrogate spaces define low-dimensional, well-behaved subsets of latent space that are (i) supported by the model, (ii) sufficiently expressive to contain high-scoring solutions, and (iii) amenable to traversal by standard optimisation algorithms. Section 5.4 then illustrates the effectiveness of this approach in a realistic setting.

### 5.1 SURROGATE SPACES ARE MODEL AND DATA-MODALITY AGNOSTIC

Our methodology acts on latent variables only, and so is agnostic to the generative model. As such, it can be applied to any model for any type of data, provided the latent variable follows the appropriate distribution or can be mapped to such a distribution via a transport map, see Section 3. We illustrate this in Figure 1 and Figure 2 with surrogate latent spaces for images, video, audio, and proteins, respectively. The formed spaces are smooth, low-dimensional, and still retain high generative quality.

### 5.2 GOOD EXAMPLES DEFINE SPACES WITH BETTER SOLUTIONS

**Goal:** This experiment isolates a key requirement of surrogate search spaces: they must preserve both sample quality and diversity to support effective optimisation.

**Setup**: We follow the benchmark of Denker et al. (2025), focusing on generation from a diffusion model to yield high scores on an image prompt-following task, while maintaining diversity. This task allows us to test if high-scoring seed solutions lead to spaces containing high-scoring solutions, and if these solutions are diverse. We evaluate the scores and diversity across grids in surrogate spaces, constructed from seeds being the top-scoring among random samples, and use state-of-the-art methods trained on the particular score function for reference.

**Results**: Table 1 reports the median and 90% confidence interval of the mean score of the 100 generated images per method over 30 repetitions. We include the reported scores for Importance Fine-tuning (Denker et al., 2025), DPOK (Fan et al., 2023b), Adjoint Matching (Domingo-Enrich et al., 2024). The mean score and diversity across the grids is high and similar to the trained methods, demonstrating that the surrogate spaces contain good and diverse solutions. Full setup and analysis is provided in Section F.

### 5.3 SURROGATE SPACES SEARCHED BY STANDARD OPTIMISATION ALGORITHMS

**Goal:** We now assess if our methodology is effective under properties typical of problems we are interested in: high-dimensional, low-success-rate tasks where optimisation is essential for finding good solutions. Moreover, we need to demonstrate that various standard optimisers can be used.

**Setup:** We construct a synthetic benchmark designed to be extremely hard to solve by sampling alone — generating a highly specific image from a very generic prompt. Using an image diffusion model supporting prompt-following allows us to construct a collection of objective functions from randomly chosen target prompts. Cheap-to-compute score functions for prompt-following exists, making it is feasible to test all combinations of multiple optimisation algorithms, search space dimensionalities, and choices of the weight chart $\phi_w$ across many runs. We use the popular methods of CMA-ES (Hansen, 2016), BO (Shahriari et al., 2015), as well as random search. The model is given a general prompt ('A vehicle') and the objective is to obtain high prompt-following scores for a prompt sampled randomly from a grammar composed of three parts as 'A <attribute> <vehicle type> <environment>', where there are 100 of each attribute, types and environments to sample from, forming one million possible combinations. The sampled target prompts are hidden from the methods, but implicitly conveyed via the objective function; the prompt-following score as a function of the generated image. At our disposal we have $M$ examples for each part where the attribute, type, or environment match the target, but where the other parts are sampled randomly. This is to simulate the scenario where the practitioner has access to informative but incomplete solutions a-priori, having some of the target characteristics, but not all. This, per setup and sampled prompt, yields a number of seeds of $K = 3M$. See setup details and full analysis in Section G.

| Method | Best RMSE ↓ | SUCC ↑ | SUCCDIV ↑ |
|---|---|---|---|
| Random in $\mathcal{Z}$ | 2.33 | 0 | 0 |
| CMA–ES in $\mathcal{U}^{K-1}$ (random seeds) | 1.19 | 7 | 4 |
| CMA–ES in $\mathcal{U}^{K-1}$ (filtered seeds) | 1.08 | 12 | 5.5 |

Table 2: Protein design at $N = 600$ after 200 iterations. SUCC: successful recoveries (RMSE$< T$); SUCCDIV: distinct clusters of successful backbones. Medians over 10 runs.

**Results:** Figure 4 shows that optimisers perform better within our surrogate spaces than in the full, original latent space ($\mathcal{Z}$), typically outperforming the best solutions found over a whole run of random search in $\mathcal{Z}$ (i.e. standard sampling from the generative model) in just handful of evaluations. CMA-ES deployed in the $\mathcal{Z}$ failed to produce anything but black images, which is expected as it is unlikely to find a point on the manifold of realistic latent realisations (see Bodin et al. (2024)). In Figure 9 we report results using surrogate spaces with an alternative choice of $\phi_w$ (see Section I), as well as all combinations of optimisers, including random search in surrogate spaces. In very low-dimensional surrogate spaces, random search in $\mathcal{U}$ was nearly as effective as BO and CMA-ES in $\mathcal{U}$, as these spaces are very simple to search within but are more limited in the solutions they contain. As the dimensionality increased (by providing more seeds) CMA-ES and BO performed substantially better, able to exploit the structure in the surrogate space to find higher scoring solutions.

### 5.4 PROTEIN OPTIMISATION WITH RFDIFFUSION

**Goal:** Finally, we demonstrate our method in a setting where no existing optimisation approach is viable: generating $N = 600$–residue proteins with RFDIFFUSION (Watson et al., 2023). In this regime, prior work almost never produces designs recoverable by ALPHAFOLD2 within the target accuracy defined by Watson et al. (2023). Here, the objective (recoverability) is expensive, gradient-based optimisation is infeasible, and the naive sample–and–filter strategy of Watson et al. (2023) remains the only practical baseline.

**Setup:** We compare standard sampling in $\mathcal{Z}$ (the pipeline of Watson et al. (2023)) with CMA–ES operating in our $\mathcal{U}$. We use both heuristically selected seeds (chosen for low RMSE) and uninformative, randomly chosen seeds. The latter tests whether performance gains genuinely require informative seeds, or whether the structure of the surrogate search space alone confers a substantial advantage.

**Results:** Table 2 and Figure 5 show that our surrogate spaces higher rates of successful recovery (RMSE$< 2.0$ Å), and a greater number of structurally diverse successful proteins. Random sampling failed in most trials, consistent with prior results. In contrast, optimisation in $\mathcal{U}$ consistently produced recoverable proteins; random seeds already improved success rates at no additional cost, while filtered seeds (taken as the 24 with lowest RMSE out of 100 random designs, as no a-priori solutions were known) further reduced RMSE and increased yield over ten-fold compared with the baseline. Full experimental details are provided in Appendix J and Appendix K.

## 6 CONCLUSION

In this paper we introduced surrogate latent spaces: a simple and general construction that enables expressive low-dimensional search spaces to be derived from high-dimensional generative models. These spaces provide structured, deterministic manifolds on which black-box optimisation becomes tractable, even under expensive or gradient-free objectives. Our experiments demonstrate that surrogate spaces offer an effective way to condition generative models via example-defined coordinates, and that their geometry is well suited to standard optimisation algorithms across modalities.

Future work will include developing techniques for predicting the utility of individual seed choices, adaptive seed-selection strategies, and integrating human-in-the-loop objective functions for creative and scientific applications.

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

## A  INVERSE OF THE $l$ MAP

In this section we will derive the inverse of the map $l$ introduced in Section 3.

**Lemma 1.** Let $\{z_k\}_{k=1}^K \subset \mathcal{Z}$ be seeds, and define their inner latents $\epsilon_k = \mathcal{T}_\rightarrow(z_k)$. Let $\boldsymbol{\xi} = [\epsilon_1, \ldots, \epsilon_K] \in \mathbb{R}^{D \times K}$. Assume:

(A1) $\boldsymbol{\xi}$ has full column rank, so that $\boldsymbol{\xi}^+ \boldsymbol{\xi} = I_K$;

(A2) For all $\epsilon \in \mathbb{R}^D$,
$$\mathcal{T}_\rightarrow\big(\mathcal{T}_\leftarrow(\epsilon)\big) = \alpha(\epsilon)\,\epsilon, \qquad \alpha(\epsilon) > 0.$$

Define the $l$ map
$$l(\boldsymbol{w}, \{z_k\}) = \mathcal{T}_\leftarrow(\boldsymbol{\xi} \boldsymbol{w}), \qquad \boldsymbol{w} \in \mathbb{S}_+^{K-1}.$$

Then $l$ is invertible with
$$l^{-1}(\boldsymbol{z}, \{z_k\}) = \frac{\boldsymbol{\xi}^+ \mathcal{T}_\rightarrow(\boldsymbol{z})}{\|\boldsymbol{\xi}^+ \mathcal{T}_\rightarrow(\boldsymbol{z})\|}.$$

*Proof.* Let $\boldsymbol{z} = l(\boldsymbol{w}, \{z_k\}) = \mathcal{T}_\leftarrow(\boldsymbol{\xi} \boldsymbol{w})$. Applying $\mathcal{T}_\rightarrow$ and using (A2) gives
$$\mathcal{T}_\rightarrow(\boldsymbol{z}) = \mathcal{T}_\rightarrow\big(\mathcal{T}_\leftarrow(\boldsymbol{\xi} \boldsymbol{w})\big) = \alpha(\boldsymbol{\xi} \boldsymbol{w})\,\boldsymbol{\xi} \boldsymbol{w}.$$

Multiplying by $\boldsymbol{\xi}^+$ and using (A1),
$$\boldsymbol{\xi}^+ \mathcal{T}_\rightarrow(\boldsymbol{z}) = \alpha(\boldsymbol{\xi} \boldsymbol{w})\,(\boldsymbol{\xi}^+ \boldsymbol{\xi})\boldsymbol{w} = \alpha(\boldsymbol{\xi} \boldsymbol{w})\,\boldsymbol{w}.$$

Thus $\boldsymbol{\xi}^+ \mathcal{T}_\rightarrow(\boldsymbol{z})$ is a positive scalar multiple of $\boldsymbol{w}$. Normalising cancels the unknown factor,
$$\frac{\boldsymbol{\xi}^+ \mathcal{T}_\rightarrow(\boldsymbol{z})}{\|\boldsymbol{\xi}^+ \mathcal{T}_\rightarrow(\boldsymbol{z})\|} = \boldsymbol{w},$$

which establishes the result. $\square$

**Corollary 1** (When normalisation is redundant)**.** Normalisation in the inverse formula is redundant if and only if
$$\mathcal{T}_\rightarrow \circ \mathcal{T}_\leftarrow = \mathrm{id},$$
that is, when $\alpha(\epsilon) \equiv 1$.

- **Gaussian latents.** $\mathcal{T}_\rightarrow$ and $\mathcal{T}_\leftarrow$ are exact inverses, so $\alpha = 1$. Normalisation is not required.

- **Hyperspherical latents.** With $\mathcal{T}_\leftarrow(\epsilon) = \epsilon/\|\epsilon\|$ and $\mathcal{T}_\rightarrow = \mathrm{id}$, one has $\alpha(\epsilon) = 1/\|\epsilon\|$. Normalisation is essential.

- **Independent scalar latents mapped via CDF to Gaussian.** Exact inverses, so $\alpha = 1$. Normalisation is redundant.

## B  $\mathbb{S}^N$ IS A SUFFICIENT INDEX FOR LATENT OPTIMAL LINEAR COMBINATIONS

In this section we will show that linear combination weights on the unit hypersphere is sufficient to index all Latent Optimal Linear combinations (Bodin et al., 2024). We will first address the Gaussian case and then the general case.

**Gaussian latents**    A linear combination

$$\boldsymbol{y} = \boldsymbol{Z}\boldsymbol{w}, \tag{8}$$

where $\boldsymbol{Z} = [\boldsymbol{z}_1, \ldots, \boldsymbol{z}_K]$, $\boldsymbol{w} \in \mathbb{R}^K$, $\boldsymbol{z}_k \in \mathbb{R}^D$, $\boldsymbol{z}_k \sim p$ and $p = \mathcal{N}(\boldsymbol{\mu}, \boldsymbol{\Sigma})$ has distribution

$$\boldsymbol{y} \sim \mathcal{N}(\alpha\boldsymbol{\mu}, \beta\boldsymbol{\Sigma}) \tag{9}$$

where $\alpha = \sum_1^K w_i$ and $\beta = \sum_1^K w_i^2$. The variable $\boldsymbol{y}$ does not follow the same distribution $p$ as $\boldsymbol{z}_k$ at weights yielding $\alpha \neq 1$ and $\beta \neq 1$. In Bodin et al. (2024) the following map was proposed for the linear combinations $\boldsymbol{y}$ in the Gaussian case

$$\mathcal{T}(\boldsymbol{y}) = (1 - \frac{\alpha}{\beta})\boldsymbol{\mu} + \frac{\boldsymbol{y}}{\sqrt{\beta}} \tag{10}$$

which is the Monge optimal map between $\mathcal{N}(\alpha\boldsymbol{\mu}, \beta\boldsymbol{\Sigma})$ and $\mathcal{N}(\boldsymbol{\mu}, \boldsymbol{\Sigma})$.

We can rewrite Equation 10 as

$$\mathcal{T}(\boldsymbol{y}) = (1 - \frac{\alpha}{\|\boldsymbol{w}\|})\boldsymbol{\mu} + \frac{\boldsymbol{y}}{\|\boldsymbol{w}\|}, \tag{11}$$

and note that if $\boldsymbol{\mu} = \boldsymbol{0}$, then the transformed variable is invariant to the norm of the weights $\boldsymbol{w}$.

As we can treat the requirement of $\boldsymbol{\mu} = \boldsymbol{0}$ by centring the distribution for a known mean vector, it follows that $\boldsymbol{w} \in \mathbb{S}^{K-1}$ is sufficient to index all such transformed variables.

**General case**    In the non-Gaussian setting, the same principle applies once we introduce an amenable inner latent distribution $p_\epsilon$ (Section 3). For any latent distribution $p$, we construct transport maps $\mathcal{T}_\rightarrow$ and $\mathcal{T}_\leftarrow$ such that $\boldsymbol{\epsilon} = \mathcal{T}_\rightarrow(\boldsymbol{z}) \sim p_\epsilon$ and $\boldsymbol{z} = \mathcal{T}_\leftarrow(\boldsymbol{\epsilon}) \sim p$. Because $p_\epsilon$ is rotationally invariant and closed under aggregation with unit–$\ell_2$ weights, any linear combination

$$\boldsymbol{\epsilon} = \boldsymbol{\xi}\boldsymbol{w}, \qquad \boldsymbol{\xi} = [\boldsymbol{\epsilon}_1, \ldots, \boldsymbol{\epsilon}_K]^T, \qquad \boldsymbol{\epsilon}_k \sim p_\epsilon, \tag{12}$$

with $\boldsymbol{w} \in \mathbb{S}^{K-1}$ again satisfies $\boldsymbol{\epsilon} \sim p_\epsilon$. Applying the inverse transport then yields

$$\boldsymbol{z} = \mathcal{T}_\leftarrow(\boldsymbol{\epsilon}) \sim p, \tag{13}$$

showing that the weights $\boldsymbol{w}$ on the unit hypersphere are sufficient to index all latent-optimal linear combinations, regardless of the underlying distribution $p$.

Concretely, the Gaussian case corresponds to the choice $p_\epsilon = \mathcal{N}(\boldsymbol{0}, \boldsymbol{\Sigma})$, where closure holds directly under $\|\boldsymbol{w}\| = 1$. For other distributions, $p_\epsilon$ and the associated transport maps adapt accordingly: hyperspherical latents are closed under normalisation, composite latents can be mapped component-wise, and scalar independent latents can be treated dimension-wise via their cumulative distribution functions. In all cases, the invariance of $p_\epsilon$ under unit–$\ell_2$ aggregation ensures that $\boldsymbol{w} \in \mathbb{S}^{K-1}$ is a sufficient index.

**Summary**    Both the Gaussian case and the general case rely on the same underlying mechanism: linear aggregation in a latent space that is invariant under unit–$\ell_2$ weighting, together with a suitable transport map back to the target distribution $p$. This establishes that restricting to $\boldsymbol{w} \in \mathbb{S}^{K-1}$ is always sufficient to represent all Latent Optimal Linear combinations, independent of the specific form of $p$.

## C    WEIGHT CHARTS $\phi_w$

Let $K$ be the number of seeds. The *weight chart* is a map

$$\phi_w : [0, 1]^{K-1} \rightarrow \mathbb{S}_+^{K-1},$$

where $\mathbb{S}_+^{K-1} = \{w \in \mathbb{R}^K : \|w\|_2 = 1, \ w_i \geq 0\}$ is the positive orthant of the unit hypersphere.

**Angular coordinates chart (spherical angles).**    Set $\theta_i = \frac{\pi}{2}u_i \in (0, \frac{\pi}{2})$ for $i = 1, \ldots, K-1$ and define

$$w_1 = \cos\theta_1, \qquad w_k = \Big(\prod_{i=1}^{k-1} \sin\theta_i\Big)\cos\theta_k \ \ (k = 2, \ldots, K-1), \qquad w_K = \prod_{i=1}^{K-1} \sin\theta_i. \tag{14}$$

**Inverse**: recover angles by $\theta_1 = \arccos(w_1)$ and $\theta_k = \arccos\big(w_k / \prod_{i=1}^{k-1} \sin\theta_i\big)$ for $k \geq 2$, then $u_i = \frac{2}{\pi}\theta_i$. **Notes**: smooth, *not* equal-area.

**Knothe–Rosenblatt (KR) chart.** Let $U \in (0,1)^{K-1}$ and define independent stick-breaks

$$v_k = I_{u_k}^{-1}\left(\tfrac{1}{2}, \tfrac{K-k}{2}\right), \quad k = 1, \ldots, K-1,$$

where $I^{-1}(a,b)$ is the inverse regularised incomplete beta. Set (Dirichlet stick-breaking)

$$z_1 = v_1, \qquad z_k = v_k \prod_{i=1}^{k-1}(1 - v_i) \ (k = 2, \ldots, K-1), \qquad z_K = \prod_{i=1}^{K-1}(1 - v_i), \qquad w_i = \sqrt{z_i}. \quad (15)$$

**Inverse**: with $z = w \odot w$ and $s_k = \sum_{j=k}^{K} z_j$,

$$v_k = \frac{z_k}{s_k}, \qquad u_k = I_{v_k}\left(\tfrac{1}{2}, \tfrac{K-k}{2}\right), \quad k = 1, \ldots, K-1.$$

**Notes**: smooth, pushes $\mathrm{Unif}([0,1]^{K-1})$ to the uniform surface measure on $\mathbb{S}_+^{K-1}$ (equal-area); use stable `betainc`/`betaincinv`.

## D THE WEIGHT CHART $\phi_w$ SETS THE SIMILARITY STRUCTURE

In Figure 6 we demonstrate numerical evidence for the claim in Section 3.2 that the dot product $\boldsymbol{w}_i^T \boldsymbol{w}_j$ is the dominant factor in determining the cosine similarity between two latent variables $\boldsymbol{\epsilon}_i, \boldsymbol{\epsilon}_j \in \mathbb{R}^D$ indexed by a surrogate latent space $\mathcal{U}$. We see that already $D \approx 100, K \leq 10$ yields an dominating $\boldsymbol{w}_i^T \boldsymbol{w}_j$, as shown by Pearson correlations of more than $0.95$, and correlations very close to $1$ for higher dimensionalities of $D$ (at a rate dependent on $K$). For reference, typical diffusion and flow matching models (Rombach et al., 2022; Labs, 2024; Lipman et al., 2022) have a $D$ of *tens of thousands* to *hundreds of thousands*, and Kong et al. (2024) has a dimensionality of several *million*.

## E EMPIRICAL STATIONARITY ASSESSMENTS OF $\phi_w$

In Section 3.3, we discuss the notion of stationarity and explain why, in practice, we must settle for approximate stationarity. Because the similarity structure is induced by the map $\boldsymbol{u} \mapsto \boldsymbol{w}$ — that is, by the weight chart $\phi_w$ — we would like our chosen $\phi_w$ to exhibit this property. In this section, we clarify what we mean by approximate stationarity in this context and present an empirical evaluation of the candidate charts for $\phi_w$ introduced in Section C. This set of candidates is not intended to be exhaustive; rather, within the scope of this work, our goal is to identify a "good enough" choice suitable for use in our methodology.

By *stationarity* we mean that the map preserves the relationship between Euclidean distances in $\mathcal{U}$ and the corresponding dot products in $\mathcal{W}$ *everywhere* in $\mathcal{U}$. Concretely, for any two pairs of points $(\boldsymbol{u}_1, \boldsymbol{u}_2)$ and $(\boldsymbol{u}_3, \boldsymbol{u}_4)$ in $\mathcal{U}$ with the same Euclidean distance,

$$\|\boldsymbol{u}_1 - \boldsymbol{u}_2\| = \|\boldsymbol{u}_3 - \boldsymbol{u}_4\|,$$

stationarity would require their mapped representations to satisfy

$$\phi_w(\boldsymbol{u}_1)^\top \phi_w(\boldsymbol{u}_2) = \phi_w(\boldsymbol{u}_3)^\top \phi_w(\boldsymbol{u}_4).$$

In other words, equal distances in $\mathcal{U}$ would correspond to equal dot products in $\mathcal{W}$. By *approximate stationarity*, we instead mean that this relationship holds only approximately: the dot products induced by $\phi_w$ are not necessarily identical for equal-distance pairs, but they remain strongly correlated with the corresponding Euclidean distances throughout most of $\mathcal{U}$.

In Figure 7, we evaluate approximate stationarity using one million pairs of points in $\mathcal{U}$, comparing both the KR map and the angular–coordinates map. In this setting, each pair is obtained by sampling two points uniformly along a randomly chosen line through $\mathcal{U}$. For comparison, Figure 8 presents the same analysis under independent uniform sampling of each point in $\mathcal{U}$. These two sampling schemes behave quite differently in high dimensions: independent sampling causes points to concentrate near the boundary of the unit hypercube, producing pairwise distances that are highly concentrated, whereas line-based sampling yields a substantially wider and more informative distribution of distances. For optimisation, the line-based scheme is more relevant, as it better reflects behaviour in all directions around typical points in the space.

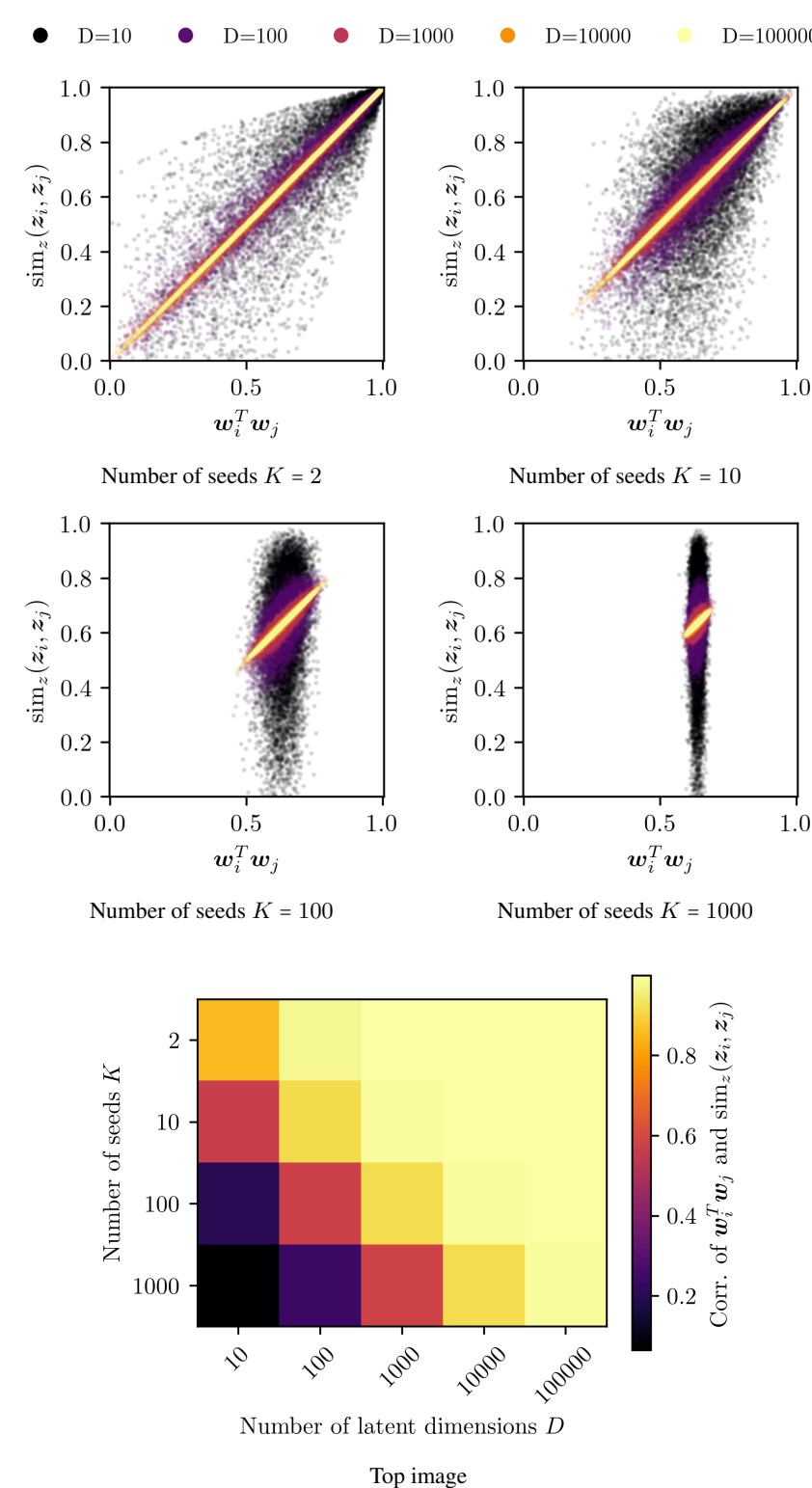

Figure 6: (*top*) Dot products of the weights $\boldsymbol{w}_i^T \boldsymbol{w}_j$ and the cosine similarity $\text{sim}_\epsilon(\boldsymbol{\epsilon}_i, \boldsymbol{\epsilon}_j)$ for uniformly drawn samples in $\mathcal{U}$ for $K = 2, 10, 100, 1000$, respectively for various dimensions $D$, where $\boldsymbol{\epsilon} \in \mathbb{R}^D$. The number of samples per setting is $10,000$, with $100$ realisations of the seeds drawn from $\mathcal{N}(\boldsymbol{0}, \boldsymbol{I})$ and $100$ uniformly sampled $\boldsymbol{u}$ per sampled seeds realisation. (*bottom*) Estimated correlations between $\boldsymbol{w}_i^T \boldsymbol{w}_j$ and $\phi_\epsilon$ uses the all the samples per setting.

Across all tested dimensionalities, the KR chart exhibits a strong and increasing correlation between Euclidean distances in $\mathcal{U}$ and the corresponding dot products in $\mathcal{W}$. In contrast, for the angular–coordinates chart, Euclidean distance becomes uninformative about the induced dot product once the dimensionality increases, with the relationship degrading rapidly as the points move further apart.

## F  GOOD EXAMPLES DEFINE SPACES WITH BETTER SOLUTIONS: DETAILS

For our surrogate spaces to be useful they need to contain varied objects that share characteristics with the seed latents; especially those attributes that impact targeted objective functions. Figure 1 presented an illustrative example, where a 2D space formed from three seeds indeed contained large areas of solutions better than the seeds. We now asses this property qualitatively on the benchmark presented in Denker et al. (2025), where we seek generations from the diffusion model Stable Diffusion 1.5 (Rombach et al., 2022) that score highly according to ImageReward (Xu et al., 2023), a measure of alignment with a target prompt. We also report diversity scores by computing one minus the mean cosine similarity of the CLIP (Radford et al., 2021) embeddings of the images.

**Results**: We test our surrogate spaces by seeing if we can form 100 generations (gridded over the surrogate space) that are both good (well-aligned with the target prompt) and diverse. To generate the seed latents that define our surrogate space, we use a budget of $S$ random generations and pick the top $K$ (a stand-in for having a-priori access to 'good' seeds). We also report the canonical baselines of taking the top 100 directly from the $S$ random generations on each run. Table 1 reports the median and 90% confidence interval of the mean score of the 100 generated images per method over 30 repetitions. To provide context for these scores, we also include the results from Denker et al. (2025) who use the same benchmarking setup to compare the performance of algorithms that require many GPU hours of training on the particular score function in order to produce generations with high scores. We include their reported scores (they only provide one repetition) for Importance Fine-tuning (Denker et al., 2025), DPOK (Fan et al., 2023b), Adjoint Matching (Domingo-Enrich et al., 2024), demonstrating that, on two out of three prompts, our surrogate spaces produce higher or same scoring generations than the expensive fine-tuning approaches, and similarly diverse. A larger relatively volume of high scoring solutions was observed for the lower dimensional surrogate spaces (but slightly less diverse) — which can be told by the grid yielding high mean scores — which is expected as the lower dimensionalities were produced from better seeds (with more random samples per seed to determine them).

## G  SURROGATE SPACES SEARCHED BY STANDARD OPTIMISATION ALGORITHMS: DETAILS

**Goal:** We will now confirm that surrogate spaces enable effective LSO in high-dimensional latent variable models by deploying popular optimisation algorithms and compare how they perform in our surrogate latent spaces against the original latent space. Our methodology enables good or informative solutions to guide the search by defining a targeted space, which is to be reflected in the test task. We use the popular methods of CMA-ES (Hansen, 2016), BO (Shahriari et al., 2015), as well as random search. The objective function is to optimise the Pick score (Kirstain et al., 2023) for generations of the Stable Diffusion (SD) 2.1 (Rombach et al., 2022) model. Specifically, the model is given a general prompt ('A vehicle') and the objective is obtain high Pick-scores for a prompt sampled randomly from a grammar composed of three parts as 'A <attribute> <vehicle type> <environment>', forming a million possible combinations. The grammar is given in Section L. The sampled target prompts are hidden from the methods, but implicitly conveyed via the objective function; the score as a function of the generated image. At our disposal we have $M$ examples for each part where the attribute, type, or environment match the target, but where the other parts are sampled randomly. This is to simulate the scenario where the practitioner has access to informative but incomplete solutions a-priori, having some of the target characteristics, but not all. This, per setup and sampled prompt, yields a number of seeds of $K = 3M$. The experiment is repeated 10 times; i.e. we sample 10 targets and their corresponding seed examples independently, and apply each optimisation algorithm and weight chart combination to these targets, producing 10 corresponding runs for which we report the median and the 90th confidence interval. For optimiser setups, see H.

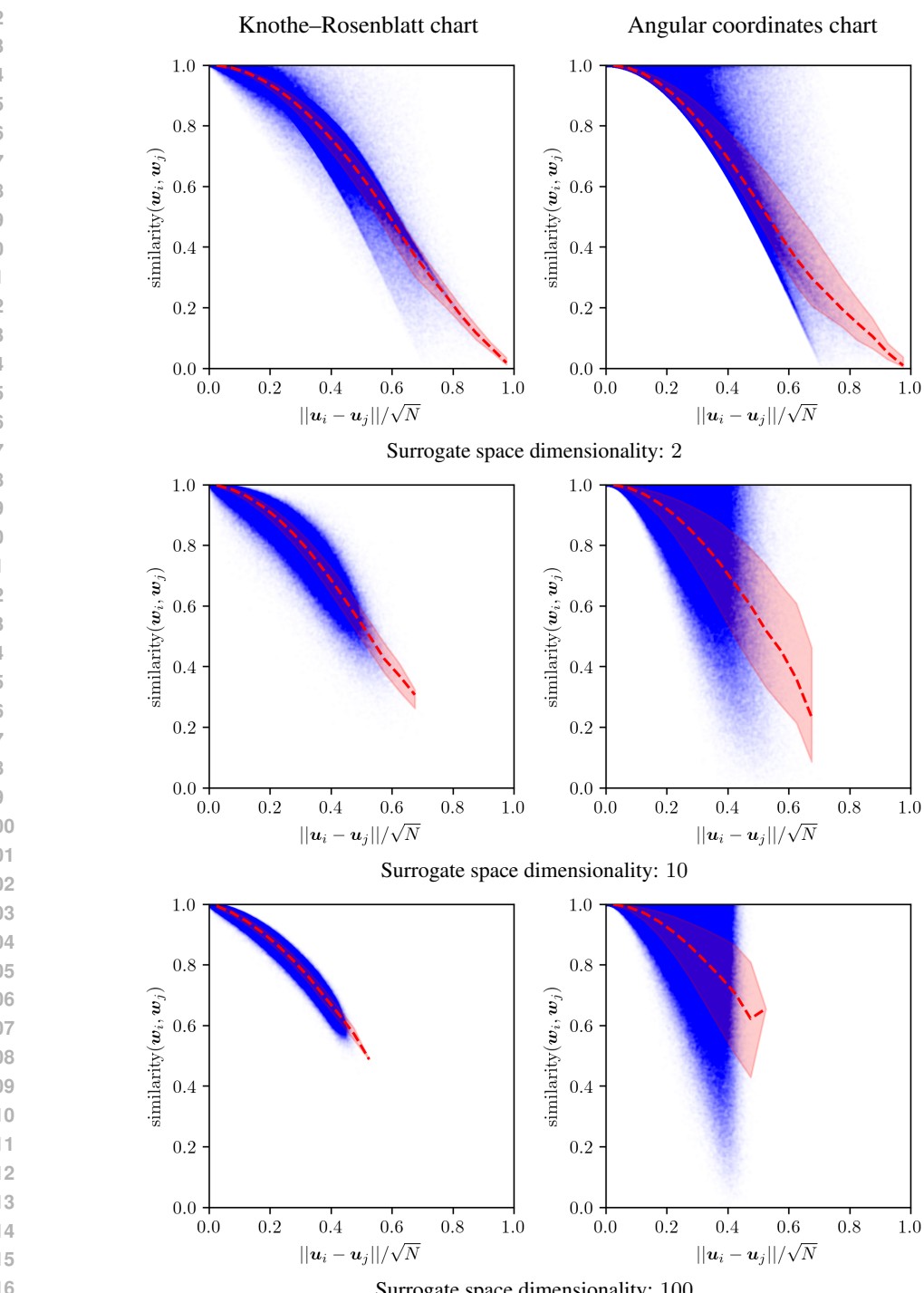

Figure 7: Shown in the blue are, for 1 million pairs of sampled points in the surrogate space ($\mathcal{U}$), their (normalised) Euclidean distance versus the similarity (dot product) of their corresponding weights in $\mathcal{W}$ (see Section 3.3) using the Knothe-Rosenblatt and Angular coordinates chart (see Section C) in the left and right column, respectively. The red dashed line shows the mean similarity and the red shared area shows the 50% confidence interval. The top, middle and bottom rows show surrogate spaces of 2, 10, and 100 dimensions, respectively, corresponding to settings of 3, 11 and 101 seeds. The point pairs have been generated through running the following procedure 1 million times: (1) sample a point $p_1$ uniformly in $\mathcal{U}$, (2) form a linear path between this point and a uniformly drawn point on the exterior of $\mathcal{U}$, (3) sample a point $p_2$ linearly between $p_1$ and the exterior point.

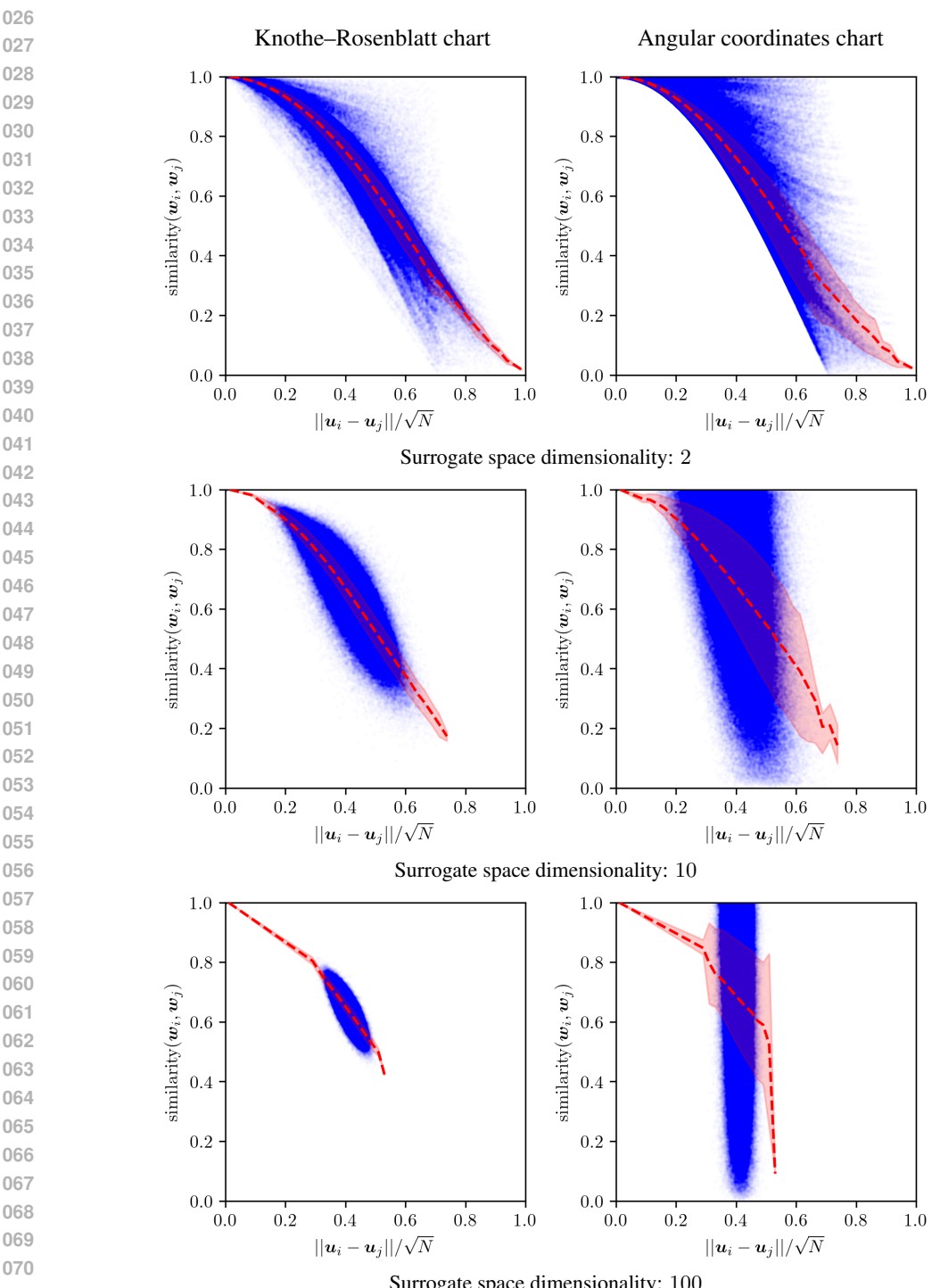

Figure 8: Shown in the blue are, for 1 million pairs of uniformly sampled points in the surrogate space ($\mathcal{U}$), their (normalised) Euclidean distance versus the similarity (dot product) of their corresponding weights in $\mathcal{W}$ (see Section 3.3) using the Knothe-Rosenblatt and Angular coordinates chart (see Section C) in the left and right column, respectively. The red dashed line shows the mean similarity and the red shared area shows the 50% confidence interval. The top, middle and bottom rows show surrogate spaces of 2, 10, and 100 dimensions, respectively, corresponding to settings of 3, 11 and 101 seeds. We note that the Knothe-Rosenblatt chart, in contrast the Angular coordinates chart, maintains a strong negative correlation between the Euclidean distance and the corresponding dot product.

**Results:** Figure 4 demonstrates that optimisers perform better within our surrogate spaces than in the full, original latent space, typically outperforming the best solutions found over a whole run of random search in the full space (i.e. standard sampling from the generative model) in just handful of evaluations. CMA-ES deployed in the original latent space (by specifying points in $\boldsymbol{u} \in [0, 1]^D$ which are subsequently mapped to latent distribution samples via the inverse Gaussian CDF) failed to produce anything but black images, which is not surprising as it is unlikely to find a point on the manifold of realistic latent realisations (see Bodin et al. (2024)). In Figure 9 we report results using surrogate spaces with alternative choices of $\phi_w$ (see Section I), as well as all combinations of optimisers, including random search in surrogate spaces. Within very low-dimensional surrogate spaces (not the full space), random search was nearly as effective as BO and CMA-ES, but as the dimensionality increased (by providing more seeds) CMA-ES and BO performed substantially better.

## H  OPTIMISER SETUPS

Optimiser setups:

- **CMA-ES.**. We use the implementation from Nomura and Shibata (2024) with population size 4 and $\sigma = 0.2$.

- **BO.** For $K = 3$ and $K = 9$ (i.e. 2D and 8D search problems), we use a Gaussian Process prior with a (3/2)-Matérn kernel and DEFER (Bodin et al., 2021) — with a budget of 300 density function evaluations and 30 hyperparameter posterior samples — for Bayesian inference for the kernel scale, lengthscale, and Gaussian (homoscedastic) noise variance parameters. For $K = 90$ (i.e. 89D search problems), we use Turbo (Eriksson et al., 2019) and the author's official implementation.

- **Random search in $\mathcal{U}$.** Uniform, independent sampling in $\mathcal{U}$.

- **Random search in $\mathcal{Z}$.** Standard random (and independent) sampling from the latent distribution.

- **CMA-ES in $\mathcal{Z}$.** CMA-ES deployed on $[0, 1]^D$, where evaluations are mapped to latent distribution samples via the inverse Gaussian CDF.

## I  WEIGHT CHART OPTIMISATION COMPARISON

In this section we report results for combinations of choices of $\phi_w$ (see Section C).

In Figure 9 we see optimisation results for two different choices of $\phi_w$; the KR and the Angular chart, respectively, in the context of each of BO, CMA-ES, and random search within the formed surrogate space, and include random search (standard sampling) in the full latent space for reference. For optimiser setups, see H. In the context of each combination of choice for $\phi_w$, optimiser, and number of seeds, the surrogate spaces substantially outperform random sampling in the full latent space. Using few seeds, i.e. low-dimensional surrogate spaces, both choices of $\phi_w$ perform similarly, while for the relatively high-dimensional surrogate space ($89D$, formed from 90 seeds) the KR chart substantially outperform the Angular chart when in the context of an optimisation algorithm (BO and CMA-ES) instead of uniform sampling within the surrogate space.

## J  RFDIFFUSION

We adopt the pipeline of Watson et al. (2023), consisting of: (1) backbone generation with RFDIFFU-SION, (2) sequence design with PROTEINMPNN (Dauparas et al., 2022), (3) structure reconstruction with ALPHAFOLD2 (Pak et al., 2023), and (4) evaluation by $C_\alpha$–frame RMSE. Lower RMSE indicates closer agreement between the generated and reconstructed backbones.

In step 1, candidate backbones are sampled from the original RFDIFFUSION model using DDIM. A backbone is defined as the set of $C_\alpha$ coordinates and residue-wise rotations, but does not include categorical amino acid identities. In step 2, each backbone is completed with $M = 8$ amino acid sequences predicted by PROTEINMPNN. This introduces the missing categorical information; however, the predictions are noisy, motivating multiple samples. In step 3, the sequences are passed to

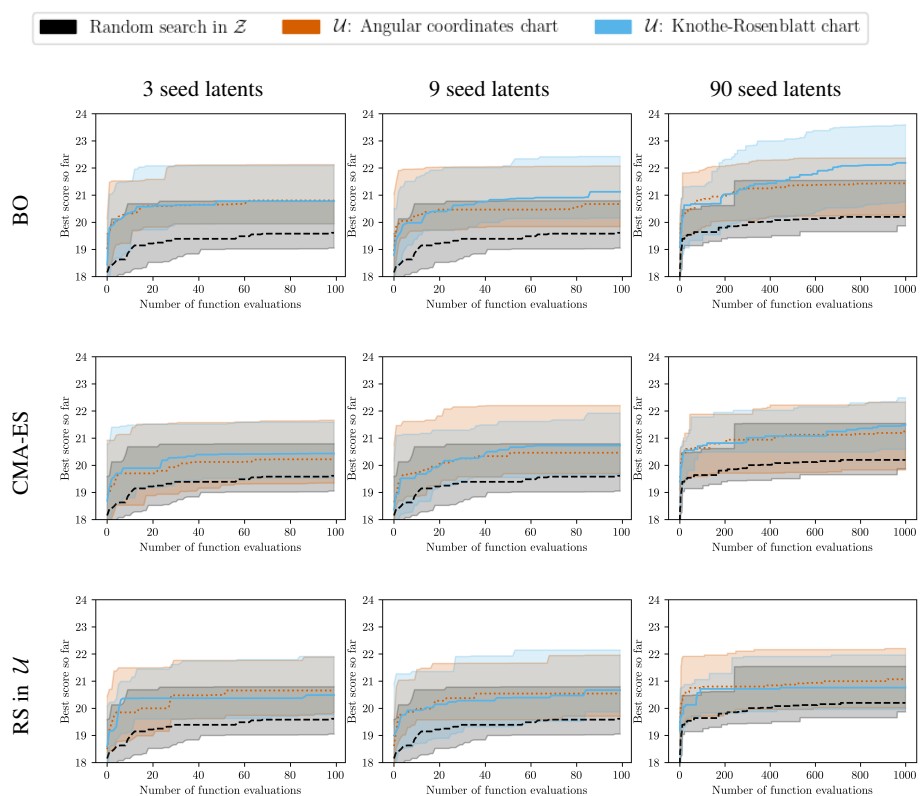

Figure 9: **Chart comparison across optimisers and surrogate space dimensionalities** Shown is the median and 90% confidence interval of the best-so-far score found per step across runs, on the task described in Figure 4 and Section G, for surrogate spaces with dimensionality $(K-1)$, where $K$ is the number of seeds (examples) provided.

ALPHAFOLD2, which reconstructs 3D structures from sequence alone, testing whether the backbone proposed by RFDIFFUSION is compatible with realistic sequences. In step 4, reconstructed proteins are aligned to the original backbones, and $C_\alpha$ RMSE is computed. For each backbone we report the best sequence (minimum RMSE over $M = 8$), following the evaluation protocol of Watson et al. (2023). For optimiser setups, see H

As in Watson et al. (2023) we adopt a threshold of $T = 2.0$ Å RMSE to define successful recovery, however we drop their secondary filtering metric of designs having PAE < 5.0 to focus on proof of principle, although in future this could naturally be supported by considering multi-objective optimisation. For fairness, all baselines were recomputed under our evaluation. Each optimisation run used 200 iterations, twice the 100 generations of the original paper.

RFDIFFUSION parametrises a backbone of length $N$ by residue-wise frames $(\boldsymbol{x}_{pos}^{(t)}, \boldsymbol{x}_{rot}^{(t)}) \in \mathbb{R}^{3N} \times \mathrm{SO}(3)^N$, where $\boldsymbol{x}_{pos}^{(t)}$ are $C_\alpha$ coordinates and $\boldsymbol{x}_{rot}^{(t)}$ are orientations derived from N–$C_\alpha$–C triplets, measured from a reference frame. The forward diffusion process applies Gaussian noise to $\boldsymbol{x}_{pos}$ and Brownian motion on $\boldsymbol{x}_{rot}$; generation is by reverse integration of the probability–flow ODE. The resulting latent is

$$\boldsymbol{z} = (\boldsymbol{z}_{pos}, \boldsymbol{z}_{rot}) = (\boldsymbol{x}_{pos}^{(T)}, \boldsymbol{x}_{rot}^{(T)}), \quad \boldsymbol{z}_{pos} \sim \mathcal{N}(\boldsymbol{0}, \mathbf{I}^{3N}), \, \boldsymbol{z}_{rot} \sim \mathrm{Unif}(\mathrm{SO}(3)^N).$$

Because $\boldsymbol{z}_{pos}$ follows a Gaussian distribution and we parametrise $\boldsymbol{z}_{rot}$ as quaternions which are uniformly distributed on $\boldsymbol{z}_{rot} \sim \mathrm{Unif}(\mathbb{S}^{3N})$, we can directly apply composite latents from Section 3.1 to construct surrogate latent spaces $\mathcal{U}$.

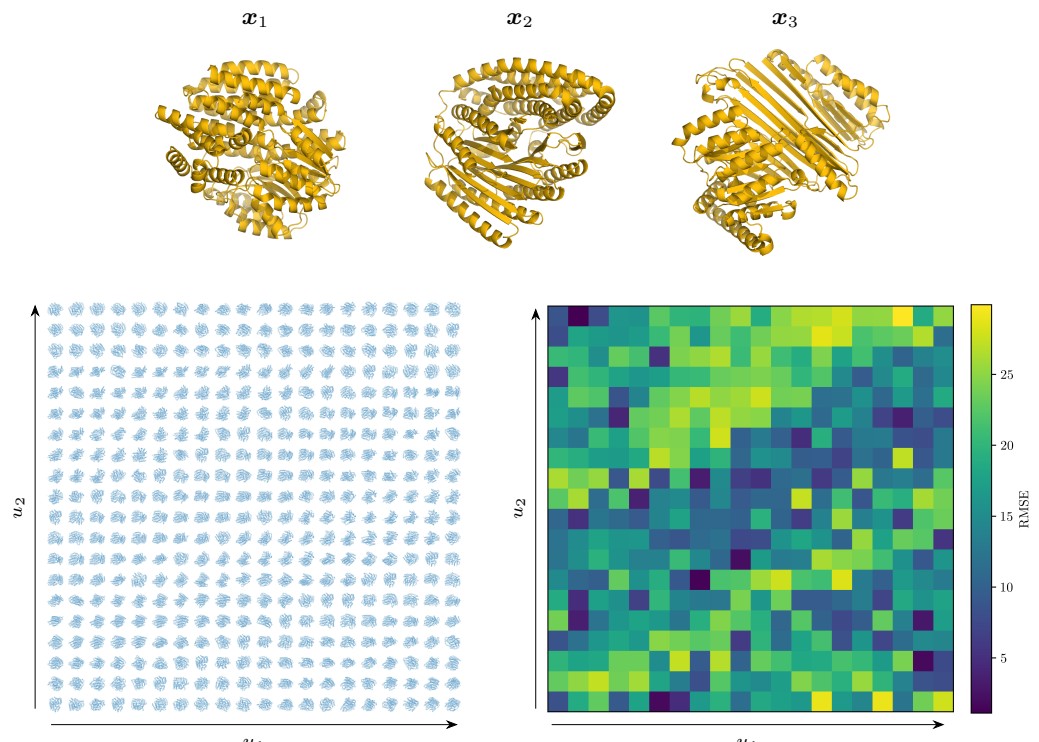

Figure 10: **2D surrogate space for proteins.** A surrogate space $\mathcal{U}^2$ defined by $K = 3$ seed latents (top) yields a structured objective landscape. *Left:* grid of generated backbones across $\mathcal{U}^2$. *Right:* corresponding evaluation scores ($C_\alpha$–frame RMSE).

Figure 10 shows a N=2 dimensional latent space formed from K=3 seed latents, over which a grid of protein structures have been generated and evaluated according to the target objective. Clear structure in shown in the objective space which makes this objective amenable to optimisation.

For the protein optimisation experiments we set the number of seed latents to $K = 24$. Optimisation is performed in $\mathcal{U}$ via CMA–ES, with candidates mapped back into $\mathcal{Z}$ for decoding and evaluation. Two seed selection strategies were used. *Random seeds:* sampled directly from the prior distributions, incurring no additional cost. *Filtered seeds:* obtained by first generating 100 backbones from the base model, ranking them by RMSE, and selecting the top $K = 24$ latents as seeds. None of these passed the $T = 2.0$ threshold, but they provided a stronger starting point than random seeds. The extra cost relative to random seeds is generating and evaluating the pipeline 100 times.

## K  TEMPLATE MODELLING SCORE (TM-SCORE)

The Template Modelling score (TM-score) is a widely used measure of structural similarity between two protein backbones. Unlike RMSE, which is sensitive to local deviations and scales poorly with chain length, the TM-score is normalised to the length of the target protein and therefore more suitable for comparing proteins of different sizes (Zhang and Skolnick, 2004).

Given a target structure of length $L$ and a comparison structure, the TM-score is defined as

$$\text{TM}-\text{score} \;=\; \max_{\text{alignments}} \frac{1}{L} \sum_{i=1}^{L} \frac{1}{1 + \left(\frac{d_i}{d_0(L)}\right)^2}, \tag{16}$$

where $d_i$ is the distance between the $i$th pair of aligned $C_\alpha$ atoms under a given alignment, and $d_0(L) = 1.24 \sqrt[3]{L - 15} - 1.8$ is a normalisation factor that accounts for protein length. The score lies in $[0, 1]$, with higher values indicating greater structural similarity.

As a rule of thumb, TM-score $> 0.5$ indicates that two structures share the same fold, while TM-score $< 0.17$ corresponds to similarity expected by chance.

In our case, all generations are of equal length, so RMSE remains valid; however, using TM-score not only allows us to apply established interpretative thresholds, but also lets us follow Watson et al. (2023) in treating two designs as *non-diverse* if their pairwise TM-score exceeds $0.6$.

**Diversity counting.** To compute the number of diverse generations reported in Section 5.4, we apply the following greedy procedure: 1. Sort generated proteins by reconstruction accuracy (lowest RMSE first). 2. Initialise the diverse set with the best structure. 3. For each subsequent protein, compute its TM-score against all members of the current diverse set. 4. Add it to the diverse set if its TM-score is $\leq 0.6$ with respect to all previously accepted members; otherwise, discard it.

This ensures that each counted generation is both accurate (passes the RMSE threshold) and structurally distinct under TM-score. *Note:* because a newly generated protein may achieve lower RMSE than existing members of the diverse set while simultaneously being non-diverse with respect to several of them, the overall count of diverse structures can decrease across iterations.

## L   IMAGE FEATURE COMPOSITION BENCHMARK GRAMMAR

Table 3 lists the possible attributes, vehicle types, and environment strings used for the prompt grammar used in the experiments reported in Section 5 (with details in Section G) and Section I.

Table 3: **Image Feature Composition Benchmark** vehicle grammar. Each combination of attributes, type, and environment strings are uniformly and independently sampled to form target prompts of the form: 'A <attributes> <type> <environment>' for the generation prompt 'A vehicle'.

| Attributes | Types | Environments |
|---|---|---|
| "red, shiny" | "sedan car" | "on a mountain road" |
| "blue, glossy" | "hatchback car" | "by the ocean beach" |
| "green, matte" | "coupe car" | "in a desert with sand dunes" |
| "black, reflective" | "convertible car" | "through a forest trail" |
| "white, clean" | "station wagon car" | "on a snowy mountain peak" |
| "silver, metallic" | "SUV" | "beside a flowing river" |
| "gold, polished" | "pickup truck" | "in a dense jungle" |
| "yellow, bright" | "minivan" | "on a frozen lake" |
| "orange, vibrant" | "cargo van" | "next to a waterfall" |
| "purple, glossy" | "limousine" | "in a grassy meadow" |
| "pink, pastel" | "sports car" | "through a rocky canyon" |
| "brown, rustic" | "microcar" | "in heavy rainstorm" |
| "gray, matte" | "standard motorcycle" | "on a wide highway" |
| "beige, plain" | "motor scooter" | "near an active volcano" |
| "teal, glossy" | "moped" | "under the northern lights" |
| "navy blue, shiny" | "dirt bike motorcycle" | "in a futuristic city" |
| "maroon, matte" | "touring motorcycle" | "inside a highway tunnel" |
| "ivory, smooth" | "cruiser motorcycle" | "on a suspension bridge" |
| "bronze, metallic" | "all-terrain vehicle (ATV)" | "beside a tall lighthouse" |
| "copper, shiny" | "utility task vehicle (UTV)" | "on a sandy dune" |
| "chrome, reflective" | "monster truck" | "in a busy marketplace" |
| "pearl white, shimmering" | "golf cart" | "under cherry blossom trees" |
| "matte black, dull" | "go-kart" | "in front of a medieval castle" |
| "glossy white, polished" | "city bus" | "at an airport runway" |
| "emerald green, shiny" | "double-decker bus" | "on a racetrack" |
| "ruby red, glossy" | "school bus" | "inside a scrapyard" |
| "sapphire blue, shiny" | "electric trolleybus" | "on a battlefield" |
| "amber yellow, glowing" | "street tram" | "in an abandoned ghost town" |
| "charcoal gray, matte" | "light rail train" | "beside a farm barn" |
| "steel silver, brushed" | "monorail train" | "through vineyards" |
| "deep purple, glossy" | "subway train" | "on cobblestone streets" |
| "forest green, matte" | "passenger train" | "in a suburban neighborhood street" |
| "sky blue, bright" | "freight train" | "beside a skyscraper" |
| "sunset orange, glowing" | "high-speed train" | "inside a factory yard" |
| "lemon yellow, bright" | "armored personnel carrier (APC)" | "on a cliffside road" |
| "rose pink, soft" | "military tank" | "through misty hills" |
| "sand beige, dusty" | "bulldozer" | "in a crater" |
| "stone gray, rough" | "excavator" | "inside a dark cave" |
| "lava red, fiery" | "forklift truck" | "in an abandoned warehouse" |
| "ice blue, frosty" | "cement mixer truck" | "under a starry night sky" |
| "neon green, glowing" | "fire engine truck" | "beside a spaceport" |
| "neon pink, glowing" | "ambulance vehicle" | "at sunset on the horizon" |
| "pastel blue, soft" | "police patrol car" | "on a frozen tundra" |
| "pastel yellow, soft" | "tow truck" | "in thick fog" |
| "midnight black, glossy" | "garbage truck" | "through rice fields" |
| "frost white, icy" | "snowplow truck" | "beside a wind farm" |
| "mirror chrome, shiny" | "logging truck" | "under a rainbow" |
| "brushed aluminum, dull" | "farm tractor" | "near a medieval stone gate" |
| "glossy teal, shiny" | "combine harvester" | "on an icy highway" |
| "metallic purple, shiny" | "horse-drawn carriage" | "in a neon-lit street" |
| "bronze, weathered" | "canoe boat" | "beside a carnival fairground" |
| "flat black, matte" | "kayak boat" | "in a junkyard" |
| "desert tan, dusty" | "rowboat" | "through a wheat field" |
| "jungle green, camo" | "pedal boat" | "in a tropical rainforest" |
| "navy gray, military" | "sailboat" | "on a wooden boardwalk" |
| "rust red, corroded" | "luxury yacht" | "at a construction site" |
| "storm gray, rough" | "catamaran boat" | "on a winding mountain pass" |
| "bright yellow, shiny" | "inflatable dinghy" | "beside a glacier" |
| "glossy red, polished" | "fishing boat" | "on a cratered moon surface" |
| "flat white, plain" | "harbor tugboat" | "inside a space station" |
| "sparkling silver, glittery" | "passenger ferry" | "through an asteroid field" |
| "dull gray, industrial" | "speedboat" | "on the surface of Mars" |
| "deep green, glossy" | "jet ski watercraft" | "inside a lunar base" |
| "ocean blue, wavy" | "hovercraft vehicle" | "inside an aircraft hangar" |
| "fire orange, glowing" | "houseboat" | "on a rocket launch pad" |
| "sun gold, shiny" | "pontoon boat" | "at a desert oasis" |
| "candy apple red, glossy" | "container cargo ship" | "on a tropical island beach" |
| "storm gray, matte" | "general cargo ship" | "in a canyon riverbed" |
| "ice silver, frosty" | "oil tanker ship" | "on an offshore oil rig" |
| "jet black, shiny" | "cruise ship" | "on a dry salt flat" |
| "steel blue, metallic" | "battleship" | "inside a military base" |
| "military green, matte" | "aircraft carrier ship" | "in an amusement park" |
| "glossy maroon, shiny" | "military submarine" | "on a snowy city street" |
| "matte navy blue" | "destroyer warship" | "beside a frozen waterfall" |
| "carbon fiber pattern" | "frigate warship" | "at a cultural festival plaza" |
| "transparent, glassy" | "hot air balloon" | "inside an industrial plant" |
| "camouflage green, patterned" | "sailplane glider" | "on a dirt trail" |
| "chrome gold, shiny" | "hang glider" | "in a foggy swamp" |
| "metallic blue, glossy" | "paraglider" | "beside a mountain lake" |
| "dark gray, dull" | "airship blimp" | "on a coastal cliff road" |
| "vibrant purple, glowing" | "helicopter" | "in front of ancient ruins" |
| "fluorescent yellow, glowing" | "gyrocopter" | "beside a pyramid" |
| "sparkling white, glittery" | "small propeller aircraft" | "inside an old temple" |
| "pearl blue, shimmering" | "seaplane" | "through rolling hills" |
| "shiny copper, metallic" | "amphibious aircraft" | "in a field of flowers" |
| "bronze, antique" | "commercial airliner jet" | "beside a farmstead" |
| "olive green, matte" | "private jet plane" | "at a roadside gas station" |
| "bright teal, glowing" | "supersonic passenger jet" | "inside a futuristic arena" |
| "plain beige, flat" | "fighter jet aircraft" | "inside a spaceship hangar" |
| "polished black, shiny" | "bomber aircraft" | "on a collapsing bridge" |
| "bright gold, reflective" | "stealth aircraft" | "on a volcanic lava plain" |
| "storm blue, dark" | "quadcopter drone" | "beside a crystal cave" |
| "camo brown, patterned" | "cargo plane" | "on a wooden pier" |
| "stealth gray, matte" | "snowmobile vehicle" | "inside a mining colony" |
| "metallic orange, glossy" | "mountain cable car" | "on a tall city rooftop" |
| "diamond white, shiny" | "space shuttle orbiter" | "through a canyon pass" |
| "rust brown, corroded" | "spaceplane vehicle" | "inside a virtual reality world" |
| "emerald green, glossy" | "rocketship" | "on an alien desert planet" |
| "jet silver, reflective" | "lunar exploration rover" | "inside a submarine base" |
| "space black, glossy" | "mars exploration rover" | "inside an underground bunker" |

