# OpenReview forum: "Define latent spaces by example: optimising over the outputs of generative models"
_ICLR.cc/2026/Conference — Submitted to ICLR 2026_

### Official Review · Reviewer_tiXi · 2025-10-21

**Soundness:** 4
**Presentation:** 3
**Contribution:** 3
**Rating:** 8
**Confidence:** 2

**Summary:**

The paper deals with sampling of generative models, aiming at identifying outputs that are probable given the estimated density but also that satisfy task-specific constraints. The authors propose to build a simple "surrogate latent space" that allows to conduct directly an optimization with regards to various tasks. The approach is evaluated in the context of text-to-image generation and protein design, exhibiting interesting results when one considers the low computation cost of the proposed approach.

**Strengths:**

* the authors adopt a constructivist approach to define their surrogate latent space, stating by establishing key principles it should support then explaining, step by step, how the space is defined and which properties they assume on the bijective functions that allow the mapping to the latent space of the model. The make a real effort of clarity, referring to the appendices for mathematical details and variations in their approach. Each time, the assumptions and their implementation are well motivated.
  - they also clearly credit previous works, in particular (Bodin et al., 2024)

* the related work is short but efficiently highlights the novelty of the proposed approach with regards to three families of approaches.

* the proposed approach is agnostic to the generative model considered (as long as it has a latent space with -- quite common -- properties) and thus the type of data handled.
  - The approach is evaluated in two very different tasks, namely two experiment in text-to-image generation and one for protein design. Each time, the approach is compared to a recent and relevant baseline.
  - The quantitative results (Table 1) are conducted over 30 repetitions and the results report the median and 90% confidence interval.

**Weaknesses:**

* the experiment on images (section 5.2) is conducted for three prompts only. However, a previous work such as (Denker et al, 2025) report their results on one prompt only. They nevertheless conduct a more ambitious experimetn on the 10 classes of MNIST (although this dataset may not be the most relevant). One must also note that the experimemnt in section 5.3 also deals with text-to-image generation and is conducted at a much larger scale. Hence, the experiment of section 5.2 can be considered as an illustrative example and is thus, in that sense, well conducted.
  - for future works (*not* rebuttal) one can suggest to conduct a similar experiment on a larger number of prompts and reporting aggregated performance. Reporting the 90% confidence interval for individual prompts is nevertheless more intersting than reporting an "average median performance and its standard deviation".

* In Table 1, the quantitative results are better for the proposed approach for two over three prompt, as highlighted, but one must consider various variant of the approach to get this result. However, the proposed approach is much less costly to compute and, in that sense, the results can be considered as intersting in comparison to the threee baselines.

* The results for protein design (Table 2) are reported in the appendix only. One can understand that there is a limited place but the main values should at least be reported in the main text, on lines 467-472.

* minor:
  - equations line 240 and 323 are not numbered
  - Snyder (1987) --> (Snyder, 1987) -- or ... problem *of* Snyder (1987)
  - although Bayesian Optimisation (BO) is defined on line 064, the full term could be used in the related works, line 341 (there is enough place)
  - references on lines 431 shoul use `\cite` instead of `\citet`
  - reference (Denker et al, 2025) line 510 reports the arxiv preprint while it has been published at Frontiers of Probabilistic Inference workshop at ICLR 2025.
  - the ICLR 2026 guideline asked for a dedicated section in the appendix explaining possible LLM usage. See "The Use of Large Language Models (LLMs)" on [this page](https://iclr.cc/Conferences/2026/AuthorGuide)

**Questions:**

Overall, the weaknesses identified are fairly minor and could be corrected by slightly reorganizing/rewriting certain sentences.

**Details Of Ethics Concerns:**

Ethic concerns are not addressed in the paper. However, there is also no obvious problem with this particular article, beyond what can be said about any article published in ICLR.

---

> ### Author Response · Authors · 2025-11-24
>
> We thank the reviewer for the very positive evaluation of the paper; finding the proposed approach novel and well motivated, the method generally applicable, and the paper clear. We address the raised points below.
>
> ### Scale of the image experiment in Section 5.2
>
> We agree that a more comprehensive evaluation would require more than three prompts. As noted, this experiment serves as an illustrative example, to demonstrate that good seeds are generally helpful and that formed surrogate spaces contain diverse solutions distinct from the seeds — it was pleasently surprising that something as simple as a grid over the surrogate space (i.e. no optimisation) can be competitive with significantly more computationally expensive (trained) methods of obtaining diverse, high-scoring samples.
>
> We have clarified the role of the experiment in the revised manuscript, and we also note the suggestion for future work on larger prompt sets.
>
> ### Variants of the method in Table 1
>
> We agree that drawing broad conclusions would require evaluation on more than three prompts. However, as discussed in the "Motivation of experiments" top level comment, the aim of this experiment is not to show that a grid over a surrogate space formed from random seeds is a strong generation method, nor to claim superiority over the baselines. Instead, the purpose is simply to demonstrate that the surrogate spaces possess the structural properties required for optimisation — namely, that they contain diverse and high-quality solutions. The naive grid within the surrogate space is intentional to illustrate this point; in practice, one would benefit from actively searching within the surrogate space for high-scoring solutions.
>
> ### Protein design results
>
> We agree that the main quantitative results from Table 2 belong in the main text. With the extended page limit for the camera-ready version these results have been moved from the appendix into the paper.
>
> ### Minor corrections
>
> We thank the reviewer for highlighting these points. In the revised manuscript:
>
> - the equations on lines 240 and 323 are now numbered;
> - the Snyder reference has been corrected to “of Snyder (1987)”;
> - “Bayesian Optimisation” is now spelled out on first use in the related works section;
> - the citations mentioned have now been fixed;
> - the Denker et al. citation has been updated to reference the published workshop version;
> - In our case, LLMs were not used for research ideation, methodological development, or substantive writing, and we interpret the instructions as not requiring an LLM-usage section in such cases:
>   “... this year, if LLMs played a significant role in research ideation and/or writing to the extent that they could be regarded as a contributor, then authors should describe the precise role of the LLM in a separate section on LLM usage.”
>   If the reviewer believes a brief statement clarifying this would still be beneficial for the camera-ready version, we are happy to add one.

---

### Official Review · Reviewer_Qm2a · 2025-10-31

**Soundness:** 2
**Presentation:** 2
**Contribution:** 3
**Rating:** 4
**Confidence:** 3

**Summary:**

This paper develops a method for optimization of deterministic generative models over arbitrary value functions. It does so by constructing a surrogate latent spacing using seed samples, which in combination with blackbox optimization procedures (like BO), recovers higher scoring samples than the seeds themselves, and in comparison to random search in latent space. The method is demonstrated on a number of modalities, including text-conditioned image generation, audio, video, and protein sequences.

**Strengths:**

- The paper tackles an important problem in optimizing arbitrary functions for deterministic-trajectory generative models with high dimensional latent space (e.g., DDIM, flow matching, etc.). The problem is clearly stated in the paper.
- The proposed solution is elegant and intuitive, appears to perform well on different modalities and without training, and the qualitative results are convincing.
- The paper is overall clearly written, though some critical sections on methodological details were somewhat dense.

**Weaknesses:**

- In general the paper may be suffering from having too many results with too little space to describe all of them in full, but critical parts are then not sufficiently described.
- This may be a lack of familiarity on my part, but section 3.1 and 3.2 could perhaps use some detail, as I was confused about the dimensionalities of the intermediate spaces and how a bijective mapping back onto the higher-dimensional latent space (z) is achieved with uniqueness guarantees. Perhaps some graphic intuition would help.
- The quantitative evaluations could be more systematic, or possibly just more clearly presented. The main result on the quality of the method is Figure 4, which, if I understand correctly, are evaluated over 1 million prompts of the same variety with the 3 different features involving vehicles in different terrains. The method clearly does better than random search, though the scoring seems arbitrary / badly scaled since a black image scores 20. More generally, it does not compare against other, more competitive methods as presented in Table 1, which seems to be only shown for the 3 prompt value functions in the different columns. Similarly, for the other modalities, while examples of the surrogate latent space and samples are given (e.g., for audio and images), I missed their quantitative evaluations, and in relation to other methods. Overall, it’s clear from the results that the proposed method works, but I’m unsure to what extent (how well relative to others) and in what settings. To be clear, I’m not saying it has to beat SOTA on every evaluation, just that it’s somewhat unclear what and how systematic the experimental setups are.
- Main result from Table 1 is essentially interpreted in one line (line 433), but there seems to be more insights to be gained by comparing across the different setups, and the table lacks highlighting or just some kind of visual guidance to help the readers make their conclusions.

**Questions:**

- What is the dimensionality of epsilon? Is the dimensionality of w = K? If so, how is the transformation to/from the lower-dimensional unit sphere orthant (w) mapped to the original higher-dimensional z- and x-space with bijective mappings (line 210-222)?
- intuitively, do the K seeds need to satisfy some kind of diversity criterion to construct a  latent space with a meaningful span? If so, how to evaluate this? Table 1 reports results on the quality and diversity of downstream generation with oversampling (e.g., best 1-of-6), but not relative to the quality / diversity of these seeds.
- are there systematic quantitative evaluations beyond the 3 example prompts in Table 1?
- Just to clarify, Fig 4 median and CI are computed over 1 million targets from the combination of the 3 attributes?
- typos on line 395 and 397? 100/500 should be 200 and 600?

---

> ### Author Response · Authors · 2025-11-24
>
> We thank the reviewer for their thoughtful and constructive assessment of the paper, and for finding the problem important, our solution elegant, and the paper clearly written.
>
> ### Methodological clarity in Sections 3.1–3.2
> We thank the reviewer for highlighting the points needing clarification. In the revised manuscript, we improved the description of the dimensionalities involved and why the mapping \(u \mapsto z\) is bijective. Importantly, the mapping is bijective only over the subset of the latent space indexed by the surrogate space — which is now clearly stated.
>
> To avoid duplication, we address the reviewer’s specific questions regarding bijectivity, dimensionality, and seed requirements in detail in our responses to your questions below.
>
> ### Clarifying the motivations for experiments
> We appreciate the reviewer’s feedback on the structure and motivation of the experimental section. Section 5 now clearly states the purpose of each experiment.
>
> The two image-optimisation tasks are designed to highlight properties that make our method effective for black-box optimisation:
> - controllable low-dimensional search spaces,
> - stable and consistent optimisation behaviour,
> - meaningful structured variation across the search space.
>
> The protein-design results then demonstrate these properties in a true black-box setting using expensive evaluators. Further details are in the “Motivation of experiments’’ section in the top level comments.
>
> Regarding scoring, we agree that PickScore exhibits largely arbitrary scaling and offset depending on the prompt; here, poor solutions score around 20, while good ones score ≥21. However, our method is evaluator-agnostic: any scoring function can be substituted without changing the optimisation process. This flexibility is a core advantage of the surrogate-space design.
>
> For audio and video, quantitative evaluation is fundamentally limited by the lack of reliable automatic metrics; prior work relies almost exclusively on human evaluations. This makes quantitative comparisons analogous to Table 1 infeasible today for these modalities. A benefit of our framework is that it increases the utility of high-quality evaluators once developed — they can be directly integrated into an optimisation loop, enabling higher-quality generation. We have clarified this point in the revised manuscript.

---

> ### Author Response · Authors · 2025-11-24
> **Questions**
>
> 1. **What is the dimensionality of \epsilon?**
>    The dimensionality of \epsilon is determined by the transport maps (Section 3.1). In most experiments we use the identity transport (appropriate for Gaussian latents), so \dim(\epsilon)=\dim(z).
>    For hyperspherical latents, the transport map projects onto the sphere, but again \dim(\epsilon)=\dim(z). In principle, with non-standard latent distributions the coordinate dimension could differ, but the *intrinsic* dimensionality (degrees of freedom) must match to preserve bijectivity.
>
> 2. **Is the dimensionality of w = K?**
>    w \in \mathbb{R}^K but is constrained to the unit sphere, i.e. w \in \mathbb{S}^{K-1}. Thus it has K-1 degrees of freedom, even though its ambient dimension is K.
>
> 3. **How is the mapping between the (K-1)-dimensional sphere and the higher-dimensional z-space bijective?**
>    The seeds define K directions in the latent space, which we collect into the matrix \xi. Given w \in \mathbb{S}^{K-1}, the map w \mapsto \epsilon is a weighted combination:
>    \epsilon = \xi w.
>    After applying the transport map (identity in most cases), this yields a smooth (K-1)-dimensional manifold inside \mathcal{Z}. If \mathrm{rank}(\xi)=K, this map is injective over the surrogate space.
>
>    The reverse map uses the pseudoinverse:
>    w = \xi^{+} \epsilon.
>    This is the exact inverse for all points generated by the surrogate space, including the seeds. It is not the inverse for arbitrary points in \mathcal{Z}; the bijection is defined only over the spanned manifold. This is now clearly described in the manuscript.
>
> 4. **Do the K seeds need to satisfy a diversity criterion? How to evaluate this?**
>    Yes, seed diversity influences the attainable variation within the surrogate space. However, this requirement is mild: Section 5.4 shows that even *random seeds* yield surrogate spaces that support effective optimisation and outperform random sampling in the full latent space.
>    Informative seeds can be used when available, but are not required; users can always add random seeds to increase coverage. A more detailed discussion appears in the “Search space construction’’ section in the top level comments.
>
> 5. **Are there systematic quantitative evaluations beyond the three prompts in Table 1?**
>    Table 1 addresses a specific diagnostic question: whether surrogate spaces contain diverse samples distinct from the seeds, which is a prerequisite for useful optimisation. To ensure comparability with prior work, we use the same three prompts introduced in earlier papers. Since this experiment is not intended to demonstrate that naive sampling from the surrogate space is itself a competitive generation method, extending it to additional prompts would not change the conclusion from the experiment. Broader quantitative validation appears elsewhere: Section 5.3 provides a substantially larger-scale text-to-image evaluation, and the protein-design experiments are the main demonstration of optimisation in a true black-box setting.
>
> 6. **Are the Fig. 4 medians and CIs computed over 1M targets?**
>    No, we compute the median and CI from 10 runs per combination of search space dimensionality, method, and function evaluation; we have now clarified this in the Appendix G. The number of possible targets are 1 million, from
> 5 which we sample the 10 targets used for these runs, so that each method and dimensionality are evaluated on the corresponding targets. The seed examples are sampled randomly per target (except sharing exactly one attribute with the target). See Figure 9 for results using additional optimisation algorithms, the Angular coordinates chart, and each of their
> combination.
>
> 7. **Typos on lines 395 and 397? Should 100/500 be 200 and 600?**
>    It is actually as intended: e.g. in the 1/2 efficiency case, a total budget of 200 samples are used to produce 100 final samples; in our case, from using 100 initial samples to select the highest scoring as seeds, then evaluating 100 samples (in a grid) within the surrogate space formed by these seeds. And analogously for the 1/6 efficiency case.

---

### Official Review · Reviewer_HMMU · 2025-11-01

**Soundness:** 3
**Presentation:** 3
**Contribution:** 2
**Rating:** 4
**Confidence:** 2

**Summary:**

The paper proposes a method to map high-dimensional representation spaces of deep generative models to low-dimensional latent spaces using example-based transformations. The approach is training-free and can be applied across various generative models for different modalities. It is qualitatively evaluated on image and protein datasets and quantitatively compared with baseline methods that require additional training, achieving comparable performance.

**Strengths:**

1. The paper addresses an interesting problem of enhancing controllability in modern generative AI models when generating new samples, which could broaden their range of applications. Considering that real-world images are often complex, defining latent spaces based on given examples is a reasonable and intuitive idea.
2. The latent spaces constructed by the proposed method satisfy desirable properties such as validity, uniqueness, and approximate stationarity.
3. The proposed method is evaluated on two distinct data modalities, and the results are promising.

**Weaknesses:**

1. Since the latent space is determined by the provided examples, it would be helpful to clarify the assumptions or requirements for selecting these examples. For instance, in Figure 4, should the examples collectively cover all key elements (e.g., matte black, dull hovercraft vehicle, vineyards) in the target image?
2. Table 1 compares several sampling strategies for the proposed method, but it remains unclear which strategy performs best or how to choose among them. Providing selection criteria or guidance would strengthen the paper.
3. According to Table 1, the proposed method does not significantly outperform baseline methods on quantitative metrics such as Reward and Diversity. While its training-free nature is a clear advantage, the reported training cost of some baselines (e.g., 4 GPU hours for Adjoint Matching) is not excessively high. Furthermore, including the inference or generation time (e.g., the average time to generate a finalized target example) would make the performance comparison more comprehensive.

**Questions:**

See "Weaknesses"

---

> ### Author Response · Authors · 2025-11-24
>
> We thank the reviewer for their thoughtful and constructive assessment of the paper, and for highlighting that our method broadens the range of applications of modern generative AI models.
>
> ### Requirements for seeds
> Please see “Search space construction’’ in the top level comments. As discussed there, the requirements on seeds depend on the difficulty of the task. The setup in Figure 4 is intentionally hard: the target image contains several distinct attributes while the prompt is extremely underspecified. In such a deliberately adversarial setting, seeds that collectively cover the key elements (e.g. colour, object, setting) make the problem solvable, whereas uninformed seeds would provide little signal.
>
> In realistic settings (such as the protein-design experiments) this is not a strict requirement: both random seeds and heuristically chosen seeds (e.g. low-RMSE structures) significantly outperform the baseline. In practice, users may select seeds informatively when possible, and can always augment them with random seeds if additional diversity is needed.
>
> The reviewer raises an important point regarding whether one can predict, *a priori*, how much a seed or set of seeds will contribute (e.g. to diversity or the attainable optimum within the formed surrogate space). This is an interesting direction for future work, and we now note this explicitly in the manuscript.
>
> ### Interpretation of image experiments
> Please see the “Motivation of experiments’’ section in the top level comments. The goal of the first image experiment (Table 1) is not to show that naively sampling the surrogate space is a state-of-the-art strategy, but to demonstrate that surrogate spaces formed from high-scoring seeds tend to be dense with high-scoring solutions, and that these solutions are distinct from both the seeds and one another. This sets the scene for effective black-box optimisation, where existing methods are not applicable.
>
> Regarding the comparison in Table 1, the relative performance of sampling strategies depends on the specific scoring function and prompt, and we avoid over-claiming general preferences. The reviewer is correct that the baselines have modest training costs in this setting; however, the strength of our method lies in its *training-free* nature, making it applicable across modalities where training comparable models are not feasible. Although the diffusion model used in this benchmark is moderately sized, retraining or fine-tuning state-of-the-art models is becoming increasingly expensive as model sizes grow. We have clarified this point in the revised manuscript.

---

### Official Review · Reviewer_DyF4 · 2025-11-09

**Soundness:** 3
**Presentation:** 4
**Contribution:** 3
**Rating:** 6
**Confidence:** 2

**Summary:**

This paper addresses the task of Latent Space Optimization (LSO) in diffusion/flow-based generative models. Authors argue that naive LSO in such models suffers from (i) Inefficient optimization due to the high dimensionality of the latent space, and (ii) difficulty to  stay on the valid support of the generative model. To address these challenges, authors propose a method for constructing lower-dimensional "surrogate latent spaces" for optimization with 3 main properties:
- **Validity** of all the latent within the space
- **Uniqueness** of the mapping between each latent in the surrogate space to the generated outputs.
- **Approximate** stationarity• of the surrogate space, where similarity between the generated objects approximately depends on the euclidean distance between their latents in the proposed space.

The proposed method constructs a small surrogate search space $U$ around a set of seed examples. To ensure uniqueness, it maps each $u\in U$ through a Weight Chart (Angular or Knothe–Rosenblatt) to a unit-norm, non-negative weight vector and uses a deterministic inversion/decoding path. To ensure validity, seed latents are combined in an inner latent via Latent Optimal Linear transport and then mapped back to the model latent. The authors argue that $U$ is approximately stationary because (i) the Weight Chart ties similarity largely to the dot product of the unit weights—hence to Euclidean distance in $U$—and (ii) in the isotropic inner latent, cosine similarity concentrates and is well-approximated by that weight dot product.

The authors evaluate their approach across multiple modalities (images, audio, video, proteins) against baselines operating in the original latent space. The authors also provide ablations and visualizations in support of the proposed method.

**Strengths:**

- The proposed method is training-free and is demonstrated across multiple modalities (images, audio, video, proteins).

- The proposed method keeps optimization on the model’s support (on-manifold), addressing a core challenge in latent-space optimization.

- Defining the search space from seed examples yields a more interpretable and steerable optimization process.

- Results
	- The method matches or exceeds fine-tuning–based baselines on the evaluated image benchmark without additional training.
	- The experiments show standard black-box optimizers (CMA-ES, BO) succeed inside the surrogate space, whereas the same optimizers often fail in the original latent (e.g., CMA-ES collapses to black images).
	- In the protein design experiments, it improves recovery metrics and the number/diversity of successful designs compared to standard sampling.

**Weaknesses:**

- Seed dependency
    - The attainable solution set is limited by the span, diversity, and quality of the chosen seed examples.
    - Each new optimization target (e.g., prompt or objective) typically requires its own seed set to construct the surrogate space.

- Approximate stationarity lacks a formal guarantee and may degrade as the surrogate dimension grows.

- Minor: The method assumes (quasi-)deterministic inversion/decoding; using stochastic samplers or strong guidance can weaken the uniqueness and approximate stationarity properties.

**Questions:**

See the weaknesses

---

> ### Author Response · Authors · 2025-11-24
>
> We thank the reviewer for the positive assessment of the method: that it addresses the core challenge of enabling Latent Space Optimisation (LSO) for diffusion and flow-matching models, and that it yields an interpretable and steerable optimisation process. Below we address the concerns raised.
>
> ### Seed dependency
> Please see “Search space construction’’ in the top level comments. It is correct that the attainable solution set is defined by the seed examples; this is an intended property of the methodology, enabling targeted search spaces. However, using seeds corresponding to meaningful or high-quality objects is not the only effective strategy. As shown in Figure 5, even forming surrogate spaces from *random seeds* provides substantial improvements over random sampling in the full latent space. See “Why random seeds work’’ in the top level comments for further explanation. Finally, the seed latents can originate from any source, and it is entirely reasonable to reuse the same seed latents across prompts or objectives, though informed task-specific choices may yield additional gains (as illustrated in Figure 5).
>
> Given the reviewer’s feedback, we have revised Sections 3 and 5 of the manuscript to ensure these points are stated clearly.
>
> ### Approximate stationarity
> We agree that the paper should provide evidence that the weight chart (KR) used in practice exhibits the approximate stationarity property described in Section 3.3 — thank you for pointing this out. Exact stationarity is impossible due to the difference in Gaussian curvature of the spaces being mapped between (Section 3.3). Formal guarantees for approximate stationarity are involved and left for future work, so we instead proceed with empirical evaluation of the specific property required.
>
> In the revised manuscript, we have added an empirical evaluation (for dimensions up to \(D = 100\); see Appendix E) showing that the KR chart (which we propose), unlike the Angular coordinates chart, preserves the relationship between Euclidean distances in \(\mathcal{U}\) and corresponding dot products in \(\mathcal{W}\) across most of the space. This is precisely the behaviour required for approximate stationarity, as discussed in Section 3.3.
>
> In Figure 9 (appendix), we show that using the KR chart for \(\phi_w\) results in better downstream optimisation performance than the Angular-coordinates chart. Developing and analysing new choices for \(\phi_w\) would be an interesting direction for future work.
>
> ### Need for deterministic inversion/decoding
> This is correct: our methodology is designed for deterministic generators, such as DDIM, the probability-flow ODE formulation of diffusion models, or flow-matching models. We state this at the beginning of Section 1. Nonetheless, and irrespective of how they were trained, deterministic generation is supported by a large proportion of available models.

---

### Author Response · Authors · 2025-11-24

We thank the reviewers for their thoughtful and constructive feedback. We are happy to see that all four reviewers find that our approach addresses an important problem; mentioning, for example, how it enables optimisers to navigate the supported manifold of diffusion and flow-matching models and broadens the applications for such models. Several reviewers also highlight the novelty of the method and the clarity of the paper.

However, several comments indicated that aspects of our motivation, and how some of the experiments relate to this motivation, were not sufficiently emphasised in the original submission. In the revised version, we make the central contribution explicit: the paper provides a practical and general framework for black-box optimisation over the outputs of diffusion and flow-matching models by constructing explicit low-dimensional search spaces. This enables any optimisation algorithm to optimise any objective function with respect to a model’s outputs, particularly for objectives that are expensive or lack gradients — a capability that did not previously exist for these classes of generative models. We also clarify the purpose of each experiment with respect to this contribution.

Finally, we have addressed all comments regarding clarity of specific sections, tables, and figures. These revisions improve readability without altering any results.

Below, we summarise responses to common reviewer comments; we refer back to these points where relevant in the individual reviewer replies.

---

> ### Author Response · Authors · 2025-11-24
> **Black-box optimisation**
>
> In black-box optimisation, objectives are often defined only through expensive simulations or experiments, and gradients are unavailable. Low-dimensional search spaces are generally essential. Prior work relied on lossy compression (e.g., VAEs), sacrificing fidelity for dimensionality reduction. Our method changes this trade-off: rather than reducing quality, we restrict search to regions of the latent space spanned by practitioner-chosen *seed* examples, preserving fidelity while narrowing the candidate set.
>
> We show that the resulting search spaces
> 1. are controlled by the chosen seed examples,
> 2. contain solutions that remain diverse and distinct from the seeds,
> 3. often include solutions significantly better than the seeds — making even *random seeds* useful when informed seeds are unavailable, and
> 4. induce structured objective/loss landscapes that standard optimisation algorithms can efficiently navigate.
>
> The dimension of the search space is set by the number of seeds, allowing it to be matched to the optimisation algorithm and compute budget.

---

> ### Author Response · Authors · 2025-11-24
> **Motivation of experiments**
>
> The protein-design task is our main demonstration that the method enables optimisation in a regime where no existing approach is viable: the objective is expensive and non-differentiable, gradient-based methods cannot be applied, and naive sampling is the only usable baseline. Our method discovers high-performing designs despite these constraints.
>
> Points (1)–(4) are validated through preceding diagnostic experiments, primarily in the image domain. Images are easy to visualise and supported by strong prompt-following models, making them ideal for isolating individual properties of the method. Since the framework is model and modality agnostic, these results transfer; Figure 2 illustrates the broad compatibility.
>
> - **Point 1:** Demonstrated first in Figure 1 through an illustrative example, then in Figure 4, showing how we can efficiently search an enormous space (of vehicles) for extremely specific objects satisfying various conditions (sampled from a million possibilities) using examples that satisfy only individual conditions.
> - **Point 2:** Table 1 isolates the requirement that search spaces must contain high-scoring and diverse solutions. Using high-scoring seeds yields spaces with substantially higher scores than random sampling, and the resulting solutions remain diverse. We include results from state-of-the-art gradient-based trained samplers only as strong reference points, not as applicable baselines for black-box settings.
> - **Point 3:** Illustrated first in Figure 1, showing (using random seed examples) that the search space often contains solutions with much higher scores than any of the seeds. This is also seen in Figures 4 and 5, where optimisation algorithms reliably find much better solutions than the seeds; both when using high-scoring and random seeds (Figure 5).
> - **Point 4:** Figures 1 and 10 show smooth and structured objective-function variation across the search space; Section 3 and the appendices detail properties such as solution validity, uniqueness, and approximate stationarity enabling efficient optimisation.
>
> Together, these results explain the success of the protein-design experiment: it is another high-dimensional optimisation problem with sparse high-value regions, but with an expensive, non-differentiable objective.

---

> ### Author Response · Authors · 2025-11-24
> **Search space construction**
>
> We clarify that *informed* seed selection (e.g. high-scoring examples) is useful but not required. Random seeds already perform strongly (Figure 5); informed seeds simply target regions of interest. As reviewers noted, diversity decreases when using fewer or less diverse seeds, but Table 1 shows that diversity remains surprisingly high even for small sets. If needed, diversity can be increased by mixing random with informed seeds.
>
> All low-dimensional search spaces involve a trade-off between optimisation difficulty and achievable optima. Figure 5 shows that this trade-off is favourable: even search spaces defined by *random seeds* yield strong improvements over random sampling in the full latent space.

---

> ### Author Response · Authors · 2025-11-24
> **Why random seeds work**
>
> Random seeds are effective because:
> - the search spaces contain solutions distinct from the seeds (Point 2 in "Motivation of experiments"),
> - these solutions can be much better than the seeds (Point 3), and
> - the induced loss/score landscape is structured enough for standard optimisation algorithms to exploit (Point 4).

---

### Meta-Review · Area_Chair_xWWF · 2025-12-24

**Summary:**

The authors introduce a framework for extracting latent embeddings from pre-trained diffusion and flow matching models, which can be used for latent-space (black-box) optimisation tasks. The latent space (U) is constructed to be a bijective mapping to the prior/noise space of diffusion and flow models (Z) which then generate designs (X, using an ODE-based method like DDIM or CFM).

This is a very interesting idea, and the reviewers all agree that it has merit.

However, the authors (and reviewers) miss a number of _key_ related works that also use diffusion models for black-box optimization tasks. These works invalidate the authors' claim of "allow[ing] black-box optimisation on top of diffusion and flow matching models for the first time". If the discussion period were allowed to continue as normal, perhaps this issue could have been addressed, but because of the prevailing circumstances, we were not afforded this opportunity, and I feel I have no choice but to recommend rejection of this paper in order to ensure the authors will address this issue in the future.

In particular, this work is not cited nor compared to and presents a latent-space BBO guidance method for diffusion models named "Latent
Multi-Objective Bayesian Optimization 2 (LaMBO-2)",

 - Gruver, N., Stanton, S., Frey, N., Rudner, T.G., Hotzel, I., Lafrance-Vanasse, J., Rajpal, A., Cho, K. and Wilson, A.G., 2023. Protein design with guided discrete diffusion. Advances in neural information processing systems (NeurIPS).

Additionally, diffusion + BBO has been used for casting high-dimensional BO as posterior inference in,

 - Yun, T., Om, K., Lee, J., Yun, S. and Park, J., 2025. Posterior inference with diffusion models for high-dimensional black-box optimization. International Conference on Machine Learning (ICML).

Also, there are a number of works on using BBO/MBO for controlled generation from diffusion/flow matching models (offline) that seem relevant:

- Yuan, Y., Chen, C., Pal, C. and Liu, X., 2025. Paretoflow: Guided flows in multi-objective optimization. ICLR
- Yao, Y., Pan, Y., Li, J., Tsang, I. and Yao, X., 2024. PROUD: PaRetO-gUided diffusion model for multi-objective generation. Machine Learning, 113(9), pp.6511-6538.

While an argument may be made that some of the aforementioned works are using task-specific generative models where pre-trained models may not exist, I believe it is still important to compare to them where possible. There have been comparisons of these against pre-trained models, e.g. such as LLMs -- though the scenario may be a stretch to use here,

 - Chen, A., Stanton, S.D., Ding, F., Alberstein, R.G., Watkins, A.M., Bonneau, R., Gligorijević, V., Cho, K. and Frey, N.C., 2025. Generalists vs. Specialists: Evaluating LLMs on Highly-Constrained Biophysical Sequence Optimization Tasks. ICML.

Some other works the authors should be aware of:
- Benchmark suite and review of HDBO methods: https://openreview.net/forum?id=RJHQAcbmpZ#discussion, https://machinelearninglifescience.github.io/hdbo_benchmark/
- A recent LSO method that may be worth comparing to: https://arxiv.org/pdf/2507.03910
- Some works on trying to formalise the problem of generative BO:
  - https://openreview.net/forum?id=1vrpdV9U3i
  - https://arxiv.org/abs/2510.25240v2

I apologise that the authors did not get a chance to address this issue, however I feel that it is crucial these works are at least referenced in the final paper, and that the authors do not claim they are the first to allow BBO to operate with diffusion and flow matching (the claim probably goes to LaMBO-2). The authors may also wish to consider benchmarking against [LaMBO-2](https://machinelearninglifescience.github.io/poli-docs/using_poli_baselines/lambo2.html) (perhaps on the [LaMBO-1 FoldX  experiments](https://machinelearninglifescience.github.io/poli-docs/using_poli/objective_repository/foldx_stability.html)) and some of the other cited generative optimization methods where possible.

**Reviewer Concerns:**

N/A in light of my concerns outlined above.

**Reviewer Scores:**

N/A in light of my concerns outlined above.

---

### Decision · Program_Chairs · 2026-01-26

Reject